# Eco-evolutionary dynamics of nested Darwinian populations and the emergence of community-level heredity

Guilhem Doulcier[1,2]\*, Amaury Lambert[3,4], Silvia De Monte[2,5†], Paul B Rainey[1,6†]

[1]Laboratoire de Génétique de l'Evolution, Chimie Biologie et Innovation, Université PSL, Paris, France; [2]Institut de Biologie de l'École Normale Supérieure (IBENS), École Normale Supérieure, Université PSL, Paris, France; [3]Laboratoire de Probabilités, Statistique et Modélisation (LPSM), Sorbonne Université, CNRS, Paris, France; [4]Center for Interdisciplinary Research in Biology (CIRB), Collège de France, Université PSL, CNRS, INSERM, Paris, France; [5]Department of Evolutionary Theory, Max Planck Institute for Evolutionary Biology, Plön, Germany; [6]Department of Microbial Population Biology, Max Planck Institute for Evolutionary Biology, Plön, Germany

**Abstract** Interactions among microbial cells can generate new chemistries and functions, but exploitation requires establishment of communities that reliably recapitulate community-level phenotypes. Using mechanistic mathematical models, we show how simple manipulations to population structure can exogenously impose Darwinian-like properties on communities. Such scaffolding causes communities to participate directly in the process of evolution by natural selection and drives the evolution of cell-level interactions to the point where, despite underlying stochasticity, derived communities give rise to offspring communities that faithfully re-establish parental phenotype. The mechanism is akin to a developmental process (*developmental correction*) that arises from density-dependent interactions among cells. Knowledge of ecological factors affecting evolution of developmental correction has implications for understanding the evolutionary origin of major egalitarian transitions, symbioses, and for top-down engineering of microbial communities.

\*For correspondence:
guilhem.doulcier@ens.fr

†These authors contributed equally to this work

## Introduction

Thirty years ago, in an article arguing the importance of the 'superorganism', Wilson and Sober expressed surprise that biologists had not recognised that communities — in the laboratory — '*could be treated as entities with heritable variation and selected accordingly*' (*Wilson and Sober, 1989*). That they might be treated as such, stemmed from recognition that the eukaryotic cell is a tight-knit community of two once free-living microbes (*Margulis, 1970*), but also from observations in nature of social insect colonies (*Wilson, 1985*), cellular slime molds (*Bonner, 1982*; *Buss, 1982*), and especially of phoretic insect communities (*Wilson and Knollenberg, 1987*).

Phoretic insect communities comprise a focal organism — often an insect such as a beetle — that moves between patchily distributed ephemeral resources carrying with it a myriad of associated organisms, including mites, nematodes and microbes. Communities associated with each insect differ by virtue of the composite members, with the conceivable possibility that some associations may harm the carrier insect, while others may bring benefit. Given that the role of dispersal is loosely analogous to a community-level reproduction event, Wilson and Sober argued that selection at the level of insect communities was likely to trump within-community selection leading to communities

'*becoming organised into an elaborate mutualistic network that protects the insect from its natural enemies, gathers food, and so on*'.

If this might happen in nature, then why might this not be realised even more potently in the laboratory? Indeed, the logic of Darwinism says it should. Provided there exists heritable variance in fitness at the level of communities, then communities will participate as units (in their own right) in the process of evolution by natural selection (*Lewontin, 1970*; *Godfrey-Smith, 2009*). In nature, the necessary conditions are likely rare (*Goodnight and Stevens, 1997*), but ecological circumstances can sometimes conspire to ensure that variation among communities is discrete, that communities replicate and that offspring communities show some resemblance to parental communities. Phoretic insect communities are a plausible case in point. In the laboratory, however, the experimenter can readily construct conditions that ensure communities (or any collective of cells) are units of selection (*Johnson and Boerlijst, 2002*; *Day et al., 2011*; *Xie and Shou, 2018*; *Xie et al., 2019*). A critical requirement is a birth-death process operating over a time scale longer than the doubling time of individual cells (*Hammerschmidt et al., 2014*; *Rainey et al., 2017*; *Black et al., 2020*).

Empirical support for the prediction that selection really can shape communities was provided by Swenson and colleagues who performed two studies in which artificial selection was imposed on microbial communities from soil (*Swenson et al., 2000a*; *Swenson et al., 2000b*). In the first, they selected communities that affected plant growth. In the second, they selected communities for ability to degrade the environmental pollutant 3-chloroaniline. In both instances, communities at extreme values of community function were repeatedly propagated. In both studies, a significant response was measured at the level of the community.

Although the finding was a surprise (*Goodnight, 2000*), it is consistent with expectations that communities of entities — no matter their identity — will participate in the process of evolution by natural selection provided communities are discrete, they replicate, and that offspring communities resemble parental communities (*Godfrey-Smith, 2009*). Discreteness is conferred by simply compartmentalising communities via their placement in independent vessels. Replication is achieved by taking a sample of the selected communities with transfer to a new vessel. Heredity, however, is less tangible, especially in the Swenson experiments, where the selected communities were pooled before redistribution into fresh vessels. Nonetheless, intuition says that heredity becomes established through interactions (*Wilson and Sober, 1989*; *Goodnight, 2000*). Understanding the mechanistic bases of community-level heredity and its emergence motivates our study.

We begin by posing a thought experiment realisable via ever improving capacity to manipulate small volumes of liquid (*Baraban et al., 2011*; *Sackmann et al., 2014*; *Cottinet et al., 2016*). Consider a millifluidic device that controls the composition of emulsions. Consider thousands of microlitre-sized droplets each harbouring communities comprised of two types of microbes that differ solely in the colour of a fluorescent protein: one type encodes a red fluorescent protein and the other a blue fluorescent protein. Interest is in the evolution of communities that are of the colour purple (an equal ratio of red-to-blue cells). Within each droplet, red and blue microbes replicate with growth rate and interaction rates being subject to evolutionary change. In the mean time, the experimenter, via lasers installed on the device, has determined the precise colour of each droplet and a priori decided that half of the droplets with composition furthest from an equal ratio of red-to-blue will be eliminated, whilst the fraction whose colour is closest to purple will be allowed to replicate. Replication involves a dilution step during which an aliquot of cells are sampled and nutrients replenished. A further round of growth then ensues along with a further round of droplet-level selection. The protocol continues thereafter with selection taking place at the level of communities via a birth-death process. In essence the schema, outlined in *Figure 1* and inherent in the work of Swenson and colleagues, involves exogenous imposition of ecological conditions sufficient to cause droplets to function as units of selection. The concept, elaborated in detail elsewhere, is referred to as 'ecological scaffolding' (*Black et al., 2020*).

Under this scaffolded-regime, communities within droplets are endowed with Darwinian-like properties (we use this term to convey the fact that removal of the scaffold leads, at least initially, to complete loss of community-level individuality). Collective-level variation is discretised by virtue of the bounds provided by the immiscibility of oil and water (communities are thus confined to droplets). Additionally, the device ensures that droplets engage in a birth-death process: droplets furthest from the collective-level trait are extinguished, whereas those closest to the colour purple are diluted and split, thus effecting collective-level reproduction. Not determined by the device however

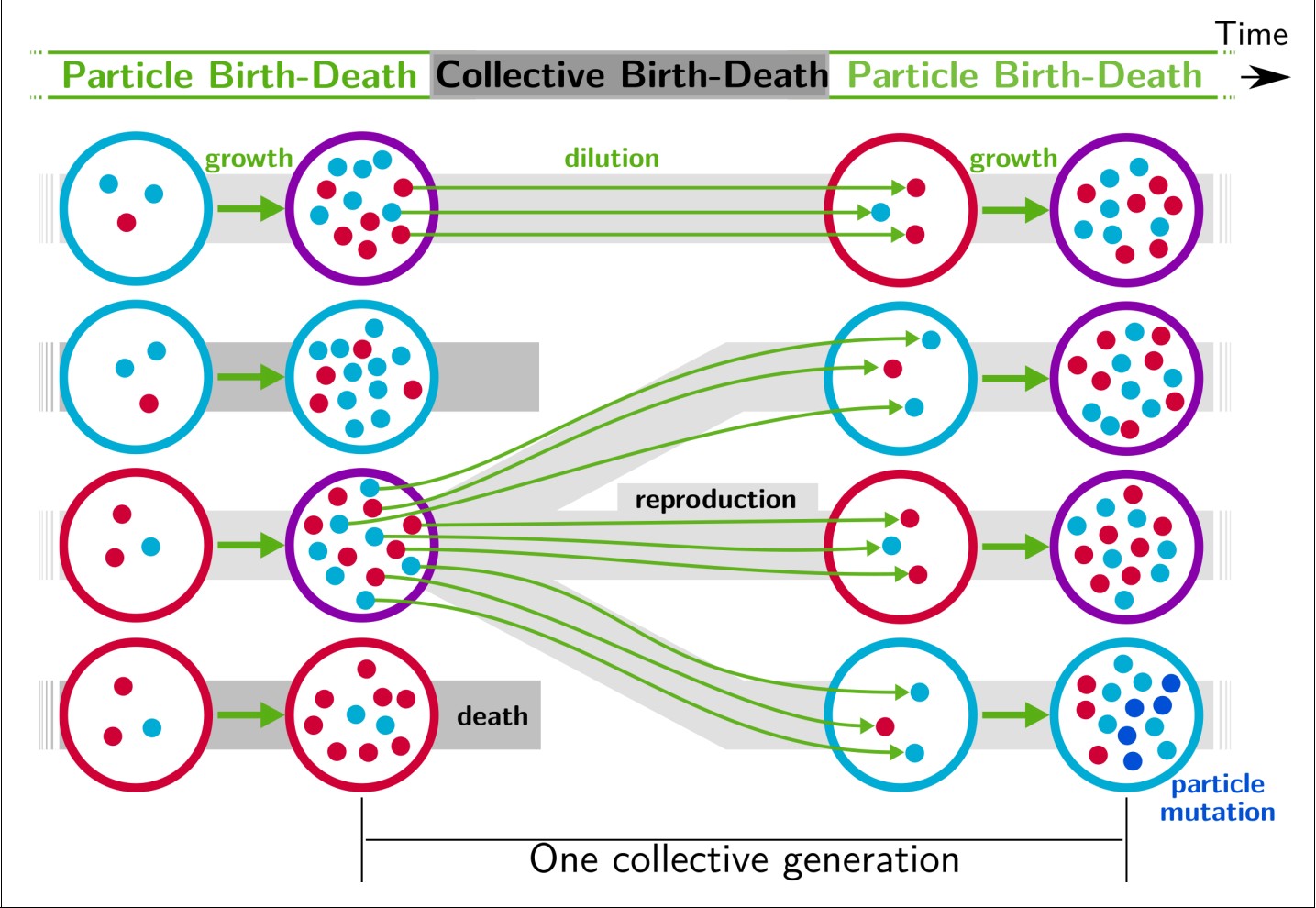

**Figure 1.** Nested model of evolution. Collectives (large circles) follow a birth-death process (grey) with non-overlapping generations. Collectives are composed of particles (small spheres) that also follow a birth-death process (*growth*, represented by thick green arrows). Offspring collectives are founded by sampling particles from parent collectives (*dilution*, represented by thin green arrows, first and third rows). Survival of collectives depends on colour. Collectives that contain too many blue (second row) or red (fourth row) particles are marked for extinction. The number of collectives is kept constant. Mutation affects particle traits (see main text for details).

is the relationship between parent and offspring droplets. Because the trait of the parent community depends on properties of the cellular constituents, there is — in the absence of interactions between red and blue cells — little chance that purple-coloured communities will reliably give rise to purple-coloured offspring. This is in part due to the stochastic nature of the dilution phase (a droplet with an equal ratio of red to blue is unlikely to give rise to offspring droplets founded with the same equal ratio of types) but also to within-droplet selection favouring fast growing types. Purple-coloured droplets can be maintained, as envisioned by the 'stochastic corrector' model (*Maynard Smith and Szathmary, 1995*; *Grey et al., 1995*; *Johnston and Jones, 2015*), provided only those communities with the correct colour are propagated. However, within-droplet selection favours rapidly growing cells resulting in successive reduction of the number of viable droplets.

Here, we show that when cellular interactions are also allowed to evolve, selection imposed at the collective level, leads to evolution of a developmental-like process, which ensures that offspring communities resemble parental communities, irrespective of the initial phenotype at the moment of birth. We illustrate the evolutionary process by means of stochastic simulations for nested populations of cells (particles) and communities (collectives) undergoing a death-birth process. In order to generalise our findings we derive a deterministic approximation, which we then use to show how

selection on community phenotype drives the evolution of ecological interactions that are the basis of community-level heredity.

## Results

### A nested model of collective evolution

As described above and depicted in *Figure 1*, we consider a nested model in which particles are discretised into a population of collectives. Each collective is comprised of two kinds of self-replicating particle (red and blue) that together determine collective colour. Colour is important because it is the phenotype upon which collectives succeed or fail. Collectives that are too far from an optimal colour face extinction, whereas those within acceptable bounds persist with the possibility of reproduction. Birth-death at the level of collectives affects the eco-evolutionary dynamics of particles as particle-level traits that give rise to unfit collectives are eliminated.

We firstly present numerical simulations where particles undergo a stochastic birth-death process (*Champagnat et al., 2006*; *Doebeli et al., 2017*) and collectives are selected based on colour. The details of the model are described in Appendix 1, but we introduce the main assumptions and underlying principles here. Each particle of type $i \in \{0, 1\}$ is characterised by four traits (hereafter *particle traits*): colour ($c_i$, red or blue), net maximum growth rate $r_i$, and two competition parameters ($a_i^{\text{intra}}$ and $a_i^{\text{inter}}$). At any particular instant particles either reproduce or die. Particles of type $i$ reproduce with a constant birth rate $r_i$ and die as a consequence of competition. The rate of death is density-dependent such that each particle of type increases the death rate of $i$-type particles by $r_i a_j^{\text{intra}}$ if they share the same colour ($c_j = c_i$), or by $r_i a_j^{\text{inter}}$ when colours are different ($c_j \neq c_i$). All transition rates can be found in Appendix 1, paragraph 'particle-level ecology'. Competition rates are referred to as 'interaction' traits or parameters. We expand on more general types of interaction — from exploitative to mutualistic — in the Discussion and in Appendix 3.

Mutations are introduced at the level of particles. Mutation affects either particle maximum growth rate ($r$) or the inter-colour competition parameter ($a^{\text{inter}}$) by a small random quantity. In the spirit of adaptive dynamics (*Geritz et al., 1998*), the particle type carrying the new set of traits is referred to as a mutant, and the existing type is designated the resident. Within every collective and at any time, there are four populations composed of resident and mutant types of the two colours. Mutations are assumed to be rare. In order to accelerate numerical simulations, one mutant individual is introduced every time one population of a given colour goes extinct in one of the collectives. The newly added type has the the same colour as the extinct type.

Collectives also undergo a birth-death process. The number of collectives $D$ is constant and collective generations are discrete and non-overlapping. Each collective generation begins at time $t = 0$ with offspring collectives containing $B$ founding particles. Particles replicate, interact and evolve according to the particle traits. After duration $T$, collectives attain 'adult' stage, and a fixed proportion of collectives $\rho$ is marked for extinction. This allows the possibility of selection on collectives based on their properties (the *collective phenotype*), which is derived from the composing particles. Our focus is collective colour, which is defined as the proportion $\phi$ of red particles.

Initially, collectives contain red and blue particles in uniformly distributed ratios. Collectives are subject to evolution under two contrasting regimes: one neutral and the other selective. Under the neutral regime, the pool of collectives marked for extinction is sampled at random, whereas under the selective regime, collectives marked for extinction are those whose adult colour departs most from an arbitrarily fixed optimal colour $\widehat{\phi}$. Extinguished collectives are replaced by offspring from uniformly sampled extant collectives (*Figure 1*). All other collectives are replaced by their own offspring. Reproduction involves uniformly sampling $B$ particles from the parent collective. Particles from one collective never mix with particles from any other. This establishes an unambiguous parent-offspring relationship (*De Monte and Rainey, 2014*). The adult colour of offspring collectives depends on the founding frequencies of particles (whose variance is negatively related to bottleneck size $B$), and on ensuing particle-level population dynamics.

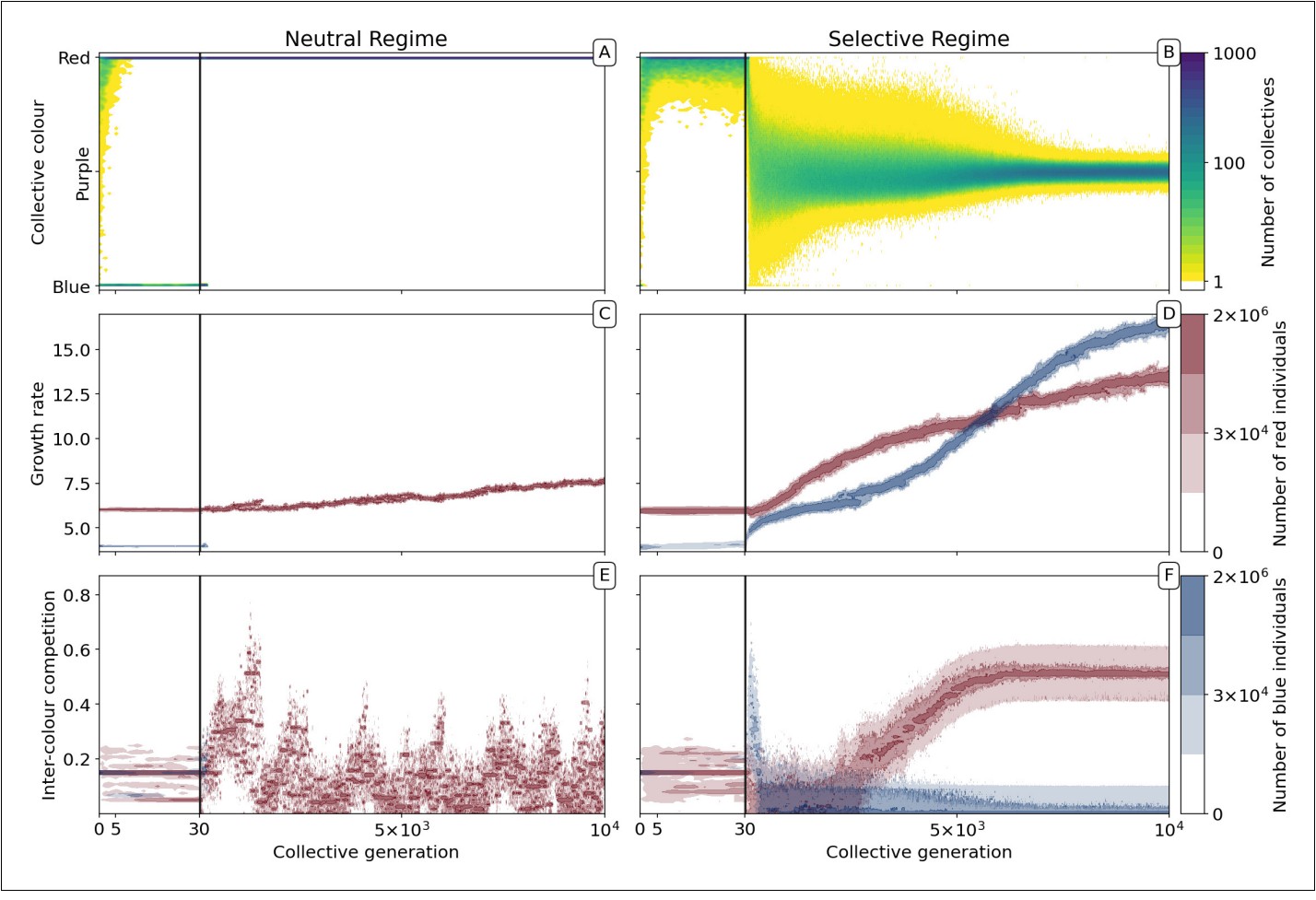

**Figure 2.** Evolutionary dynamics of collectives and particles. A population of $D$ = 1000 D collectives was allowed to evolve for $M$ = 10,000 generations under the stochastic birth-death model described in the main text (see Appendix 1 for details on the algorithm used for the numerical simulations). Initially, each collective was composed of $B$ = 15 particles of two types: red ($r_0 = 6, a_0^{\text{intra}} = 0.8/K, a_0^{\text{inter}} = 0.15/K, c_0 = \text{red}$) and blue ($r_1 = 4, a_1^{\text{intra}} = 0.3/K, a_1^{\text{inter}} = 0.15/K, c_1 = \text{blue}$), with $K$ = 1500. The proportions at generation 0 were randomly drawn from a uniform distribution. At the beginning of every successive collective generation, each offspring collective was seeded with founding particles sampled from its parent. Particles were then grown for a duration of $T$ = 1. When the adult stage was attained, 200 collectives ($\rho$ = 20%) were extinguished, allowing opportunity for extant collectives to reproduce. Collectives were marked for extinction either uniformly at random (*neutral regime*, panels A, C, E, as well as *Appendix 1—figures 1A* and *4A*), or based on departure of the adult colour from the optimal purple colour ($\widehat{\phi} = 0.5$) (*selective regime*, panels B, D, F, as well as *Appendix 1—figures 1B* and *4B*). Panels A and B, respectively, show how the distribution of the collective phenotype changes in the absence and presence of selection on collective colour. The first 30 collective generations (before the grey line) are magnified in order to make apparent early rapid changes. In the absence of collective-level selection purple collectives are lost in fewer than 10 generations leaving only red collectives (A) whereas purple collectives are maintained in the selective regime (B). Panels C-F illustrate time-resolved variation in the distribution of underlying particle traits. A diversity of traits is maintained in the population because every lineage harbours two sets of traits for every colour (see Appendix 1). Selection for purple-coloured collectives drives evolutionary increase in particle growth rate (D) compared to the neutral regime (C). In the neutral regime, inter-colour evolution of competition traits is driven by drift (E), whereas with collective-level selection density-dependent interaction rates between particles of different colours rapidly achieve evolutionarily stable values, with one colour loosing its density-dependence on the other (F).

## Selection on collectives drives the evolution of particle traits

In the absence of collective-level selection (neutral regime), collectives converge to a monochromatic phenotype (*Figure 2A*). Once collectives are composed of either all-red or all-blue particles, the contrasting colour cannot be rescued (colour change by mutation or migration is not possible). The distribution of collective colour becomes biased toward faster-growing particle types, with selection driving a gradual increase in particle growth rate (*Figure 2C*). The inter-colour competition trait

(*Figure 2E*) is irrelevant once collectives become monochromatic (evolution is then governed by drift).

The dynamic is very different once selection is imposed at the level of collectives. By rewarding collectives closest to the colour purple (a fixed $\widehat{\phi} = 0.5$ ratio of red to blue particles), it is possible to prevent fixation of either colour (*Figure 2B*). Starting, as above, from collectives containing red and blue particles in uniformly distributed ratios, mean collective colour shifts toward red. The time scale is, as in the neutral case, a consequence of the faster initial growth rate of red particles. For a few tens of generations, the population of collectives remains strongly biased towards red. The optimal phenotype is maintained by selection for the least worse collective colour precisely as envisaged by the stochastic corrector model (*Maynard Smith and Szathmary, 1995*; *Appendix 1—figure 5*). Subsequently, however, the trend reverses and mean collective colour progressively approaches purple. From generation 1000, variance in the distribution of colour decreases, as a consequence of improvement in the ability of purple-parent collectives to give rise to offspring collectives that at adult age resemble parental types. This is associated with escalating particle growth rate (*Figure 2D*) and a saturating increase in between-colour competition (*Figure 2F*). The latter reflects directional selection that moves the average phenotype in the population of collectives towards the optimal colour $\widehat{\phi}$ (reached by generation 7000).

By affecting particle traits, selection on colour also modifies dynamics within collectives. *Figure 3* shows variation of colour within a single collective growth phase at generation 3 and generation 9000. Prior to selection shaping particle traits, both red and blue particle types follow approximately exponential growth (*Figure 3C*). The resulting adult collective colour is thus biased towards the faster-growing red type (*Figure 3A*). In contrast, at generation 9000 (*Figure 3B*), both particle types reach a saturating steady state that ensures that adult colour is purple. Initial departures from a 1:1 ratio — caused by the stochasticity of collective reproduction and/or particle growth dynamics —

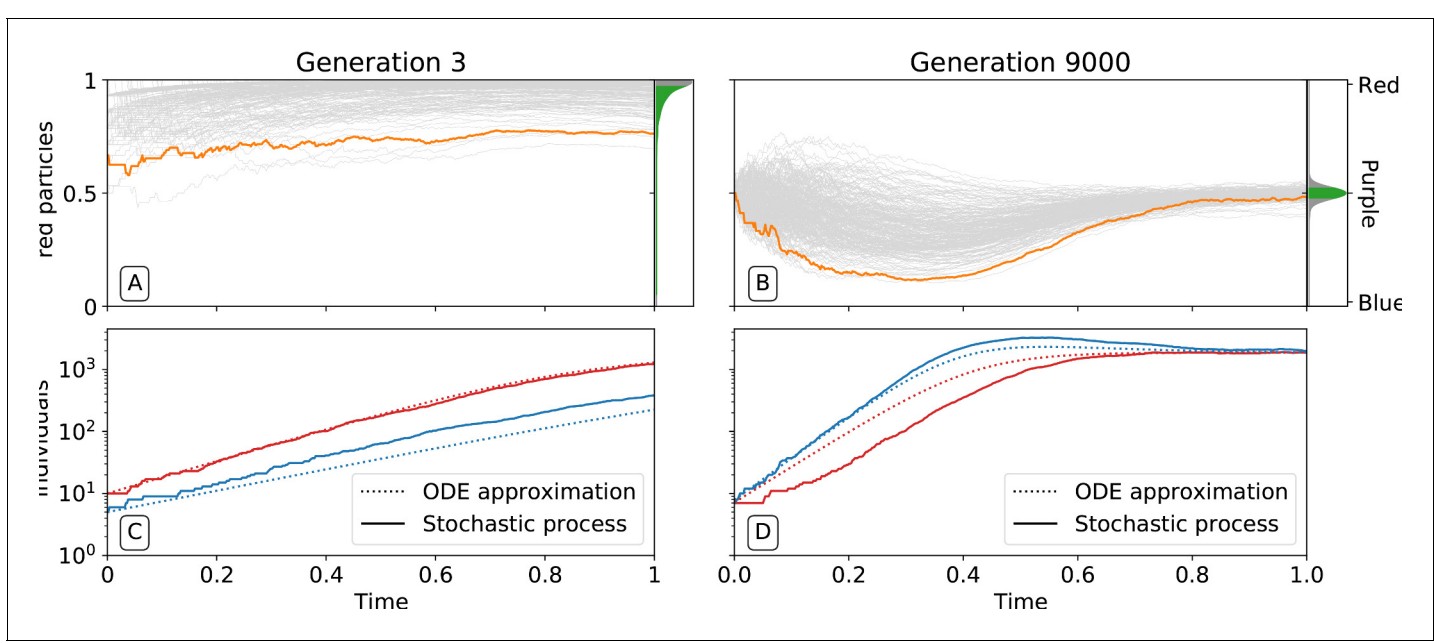

**Figure 3.** Ecological dynamics of particles. A sample of 300 (from a total of 1000) collectives were taken from each of generations 3 (A,C) and 9000 (B, D) in the evolutionary trajectory of *Figure 2*. The dynamic of particles was simulated through a single collective generation ($0 \leq t \leq T = 1$), based on the particle traits of each collective. Each grey line denotes a single collective. The frequency distribution of adult collective colour (the fraction of red particles at time $T$), is represented in the panel to the right. The grey area indicates the fraction $\rho$ of collectives whose adult colour is furthest from $\widehat{\phi} = 0.5$, that will be eliminated in the following collective generation. Single orange lines indicate collectives whose growth dynamic — number of individual particles — is shown in C and D, respectively. Dotted lines show the deterministic approximation of the particle numbers during growth (Appendix 2 *Equation 1*). Initial trait values result in exponential growth of particles (C), leading to a systematic bias in collective colour towards fast growing types (A). Derived trait values after selection yield a saturating growth toward an equilibrium (B) leading to the re-establishment of the purple colour by the end of the generation, despite initial departure (A). This is associated to the transition from a skewed distribution of collective colour, where almost all collectives are equally bad, to a narrow distribution centered on the target colour.

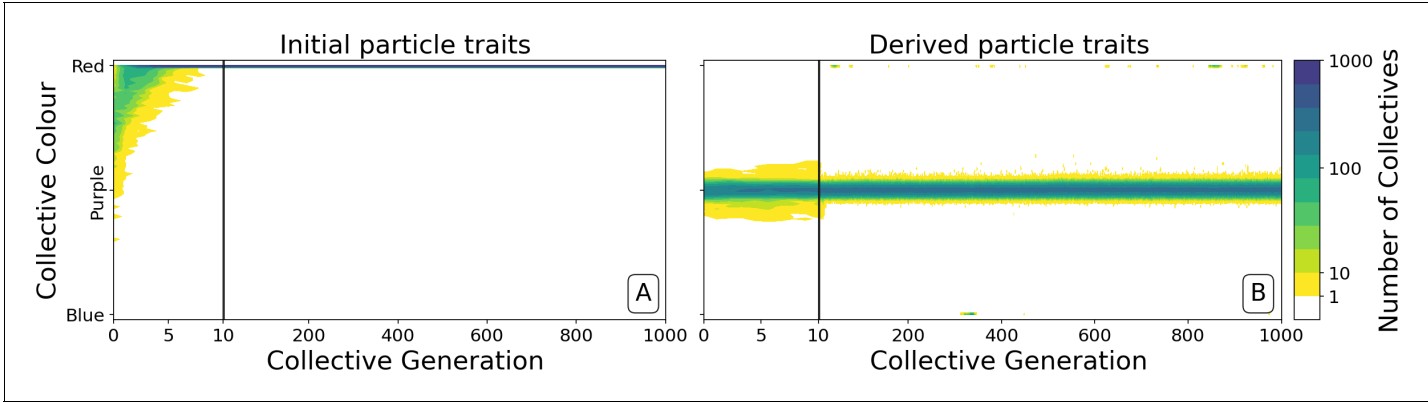

**Figure 4.** Dynamics of ancestral and derived collectives in the neutral regime. Comparison of the dynamics of the colour distribution after removing selection (neutral regime). The population of 1000 collectives is initially composed of collectives with a colour distribution identical to that at generation 10,000 in *Figure 2B*. Particle traits are: (A) as in generation 1 of *Figure 2*; (B) derived after 10,000 generations of collective-level selection for purple. In both instances, particle mutation was turned off in order to focus on ecological dynamics, otherwise parameters are the same as in *Figure 2A*. *Appendix 1—figure 6* shows the outcome with particle mutation turned on. The first 10 collective generations are magnified in order to make apparent the initial rapid changes. The particle traits derived after evolution are such that the majority of collectives maintains a composition close to the optimum $\widehat{\phi}$ even when the selective pressure is removed. This feature is instead rapidly lost in populations of collectives with the same initial colour, but with particle traits not tuned by evolution.

are compensated for during the growth phase (*Figure 3D*). Compensation is a consequence of the evolution of inter-colour competition traits (*Figure 2F*). Population expansion is in turn dependent upon earlier increases in particle growth rate (*Figure 2D*). Moreover, selection favours competition trait values for which blue types have no effect on red types: $a^{inter}$ of blue types is close to zero by generation 5000 (*Figure 2F*).

Key features of the evolutionary trajectory discussed so far are representative of replicate realisations of the stochastic individual-based nested model. In the section 'Variability of the derived particle traits' of Appendix 1 we show that in repeated simulations, the average adult collective colour always falls within a few percentage of the target colour. Moreover, collective-level selection applies more stringently to interaction parameters than to growth rates, with the latter showing a broader distribution of derived values. These conclusions also hold when the target colour $\widehat{\phi}$ is different from 0.5, with broader variation from one realisation to the other and slower convergence to the target as colour ratios become more extreme (see Appendix 1, paragraph 'Different target colours').

Ability of offspring collectives to correct departures from the optimal colour during the course of growth is akin to a developmental, or canalising process: irrespective of the phenotype of the newborn (which will likely be different to that of the adult) the child — as it grows to adulthood — develops a phenotype that closely resembles that of the parent. Evidence of this apparent canalising process can be seen upon removal of collective-level selection (*Figure 4*). Collectives founded by particles with ancestral traits become composed of a single (red or blue) colour in less than 10 generations (*Figure 4A*). In contrast, derived collectives are comprised of particles whose traits ensure that collectives continue to express phenotypes narrowly distributed around the optimal (purple) phenotype (as long as there is no mutation [*Figure 4B*]). Even when mutation is allowed to drive within- and between-collective dynamics, stability of phenotype holds for more than 200 generations (*Appendix 1—figure 6*).

## From particle ecology to collective phenotype

To understand the mechanistic basis of the canalising process, particle traits must be linked to the evolutionary emergence of collective-level inheritance, which we define as the capacity of collectives to re-establish the parental collective colour. *Figure 5* shows the relationship between the initial colour of collectives at the moment of birth (the moment immediately following dilution, $t = 0$ [the newborn colour]), and collective colour after a single particle growth cycle (the moment immediately preceding dilution, $t = T$ [the adult colour]). *Figure 5A* shows this relationship at generation 3 while *Figure 5B* shows this relationship at generation 9000.

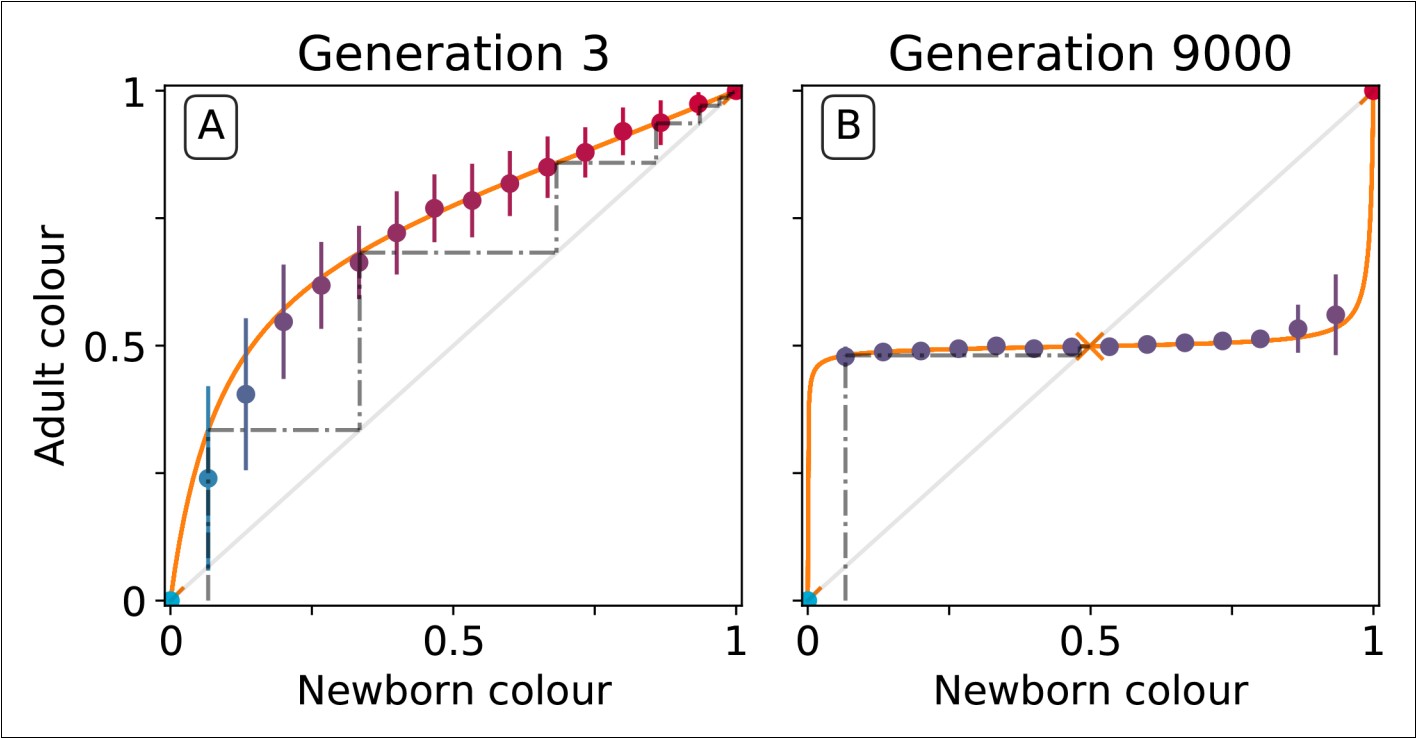

**Figure 5.** Effect of collective-level selection on newborn-to-adult colour. The adult colour of collectives as a function of their newborn colour is displayed for collectives of uniformly distributed initial colour. Stochastic simulations are realized by using particle traits representative of: (A) generation 3 and (B) generation 9000 (as in *Figure 3*). Dots indicate the mean adult colour from 50 simulations and its standard deviation. The orange line depicts the growth function $G$ for the corresponding deterministic approximation (see main text and Appendix 2). The dashed line traces the discrete-time deterministic dynamics of the collective colour, starting from $\phi = \frac{1}{B}$, and across cycles of growth and noise-less dilution. For ancestral particle traits (A), collective colour converges towards the red monochromatic fixed point. After selection for collective colour (B), the growth function is such that the optimum colour $(\widehat{\phi})$ is reliably produced within a single generation for virtually the whole range of possible founding colour ratios. The latter mechanism ensures efficient correction of alea occurring at birth and during development.

At generation 3, the proportion of red particles increases (within a collective generation), irrespective of the initial proportion. This is because red particles grow faster than blue and the primary determinant of particle success is growth rate (interactions are negligible in exponential growth). Thus, the only way that purple collectives can be maintained is if the collective phenotype is sufficiently noisy, to ensure that some collectives happen by chance to be purple, due to, for example, stochastic effects at dilution. Even if offspring collectives do not resemble their parents, purple colour is maintained via strong purifying selection that purges collectives that are either too red or too blue. The mechanism is stochastic correction (*Maynard Smith and Szathmary, 1995*; *Grey et al., 1995*; *Johnston and Jones, 2015*).

This is in marked contrast to the situation at generation 9000. After a single growth cycle, the proportion of red particles increases when the initial proportion is below, and decreases when it is above, the optimal proportion 0.5. Thus, at generation 9000, irrespective of initial conditions, the adult colour of any given collective will be closer to $\widehat{\phi} = 0.5$ than it was on founding. Accordingly, extreme purifying selection is no longer required to maintain the parent-offspring relationship. Indeed, offspring collectives return to the parent phenotype even when the phenotype at birth departs significantly from the parent (adult) phenotype. "Correction" stems from the ecological dynamics of the particles and resembles a developmental process. Hereafter we refer to this correction process as the *developmental corrector*.

The relationship between newborn and adult colour of collectives shown in *Figure 5* can be used to follow the fate of collectives over several cycles of growth and reproduction, provided the stochastic effects associated with the dilution phase are momentarily ignored. The iteration using particle trait values from generation 3 is shown by the dotted line in *Figure 5A* (the adult colour of a

collective is the newborn colour for the next cycle, following a 'staircase' geometric procedure). Because red particles grow faster than blue, it takes just six collective generations for red particles to fix within collectives. Conversely, after particle trait evolution (*Figure 5B*), the same staircase approach applied to newborn collectives of any colour shows rapid convergence to the colour purple (0.5) irrespective of the starting point. The difference in the relationship between initial and final colour at generation 3 and 9000 is evidence of the emergence of a mechanism for developmental correction.

In order to systematically explore the possible newborn-to-adult colour map and to understand how it changes through the evolution of particle traits, we use a deterministic approximation (orange line in *Figure 5*). This approximation is denoted $G$ or *growth function* (Appendix 2, Definition 2) and stems from an ordinary differential equation model often referred to as the competitive Lotka-Volterra system (Appendix 2, *Equation 1*). This model is the limit for vanishing noise of the stochastic particle ecology, and provides a good approximation of the simulations (Dotted lines in *Figure 3C–D*). The growth function $G$ captures the outcome of the ecological dynamics (i.e. the fraction of red particles) after founding populations are allowed to grow for a finite time interval $T$. We note similarity between the $G$ function and the recently proposed 'community-function landscape' (*Xie and Shou, 2018*). The shape of $G$ depends on the value of particle traits $\theta$ (growth rates $r_0$ and $r_1$, and competition parameters $a_{00} = a_0^{\mathrm{intra}}$, $a_{10} = a_0^{\mathrm{inter}}$, $a_{01} = a_1^{\mathrm{inter}}$, $a_{11} = a_1^{\mathrm{intra}}$), but also on the bottleneck size at dilution $B$ and the collective generation duration $T$. The fixed points of $G$ (i.e. $\phi$ such that $G(\phi) = \phi$) are of particular interest: in the deterministic model, these represent colours that are left unchanged during a generation. Such a fixed point is stable if the colours of collectives starting in its neighbourhood all converge to it ($\phi = 1$ in *Figure 5A*, $\phi = 0.5$ in *Figure 5B*), and unstable otherwise ($\phi = 0$ in *Figure 5A*, $\phi = 0$ and $\phi = 1$ in *Figure 5B*).

Under collective-level selection for colour, $T$ and $B$ are constant and particle traits evolve so that $G$ eventually has a stable fixed point, corresponding to the target colour $\widehat{\phi}$. Progressive change in shape of the $G$ function across collective generations in a simulated lineage (*Figure 2B*) is illustrated

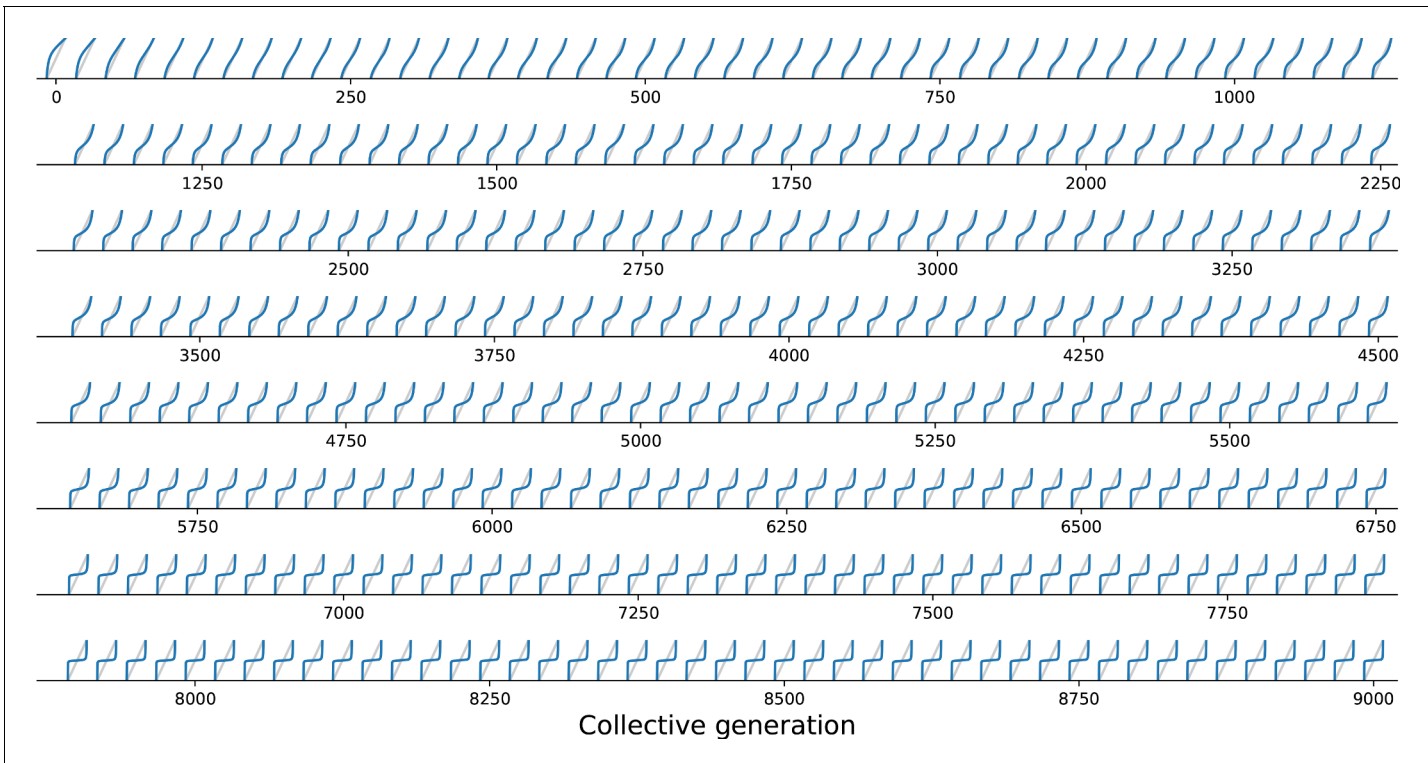

**Figure 6.** Evolutionary variation of the growth function under collective selection. function associated with the resident types for a single lineage of collectives from the simulation of *Figure 2B*, plotted every 20 collective generations from 0 to 9000. The result of iterations of $G$ gradually changes from fixation of the fast growing particle (*Figure 5A*) to convergence toward the colour purple (*Figure 5B*).

in *Figure 6*. Note that these changes are continuous: small mutations in particle traits reflect as small changes in the shape of $G$.

The evolutionary trajectory of *Figure 2B* can now be understood in terms of the progressive evolution of particle traits (see Appendix 2 for a detailed description). At the beginning, particles compete mostly during exponential phase, so that adult colour is biased towards the fast-growing type. Initial improvement in transmission of colour from parent to offspring arises as exponential growth rates $r_i$ of the particles align. Correspondingly, the $G$ function approaches linearity. A successive increase in maximal growth rate separates particle and collective time scales, allowing particles to experience density-dependent interactions. Eventually, such interactions evolve towards a regime where the $G$ function is nonlinear and fluctuations are readily compensated, thus developmental correction ensures a reliable colour inheritance.

The $G$ function, which allows characterisation of particle ecology, can now be used as a guide to optimise the 'life cycle' of growth and dilution that acts as a scaffold for the evolutionary process. In a typical experiment of community-level evolution, collective generation duration $T$ and bottleneck size $B$ are fixed. Some choices of these collective-level parameters are however likely to lead to collective phenotypes that are so far from the optimum that collective lineages go extinct. For instance, if in the first-generation competitive exclusion occurs rapidly, then distinguishing collectives based on collective colour may be impossible. Intuition suggests that the closer the fixed point of the $G$ function is to the target colour, the more efficient collective-level selection will be, and the faster the evolutionary dynamic. It is thus possible to use the distance between the fixed point of $G$ and the target composition $\widehat{\phi}$ as a proxy for the probability that collective lineages will go extinct before attaining the desired colour. Below, we examine how the position of the fixed point of $G$ changes as a function of collective generation duration $T$ and bottleneck size $B$.

## Effect of collective generation duration and bottleneck size

The growth function $G$ is readily computed from the particle traits and collective parameters even though it has in general no analytic expression (but see Appendix 2 for limit cases of exponential and saturating particle growth). There are four possible qualitative shapes of $G$, that differ in the position and stability of the fixed points (illustrated in *Appendix 2—figure 3-1 to 3-4*).

The qualitative dependence of $G$ and its fixed points on collective-level parameters varies with the underpinning particle ecology, making it easier for some communities to be starting points for the successful evolution of inheritance. Particle traits can be classified in four broad classes, based of the nature of the corresponding ecological equilibrium. For each of these classes, and when red particles grow faster than blue $r_0 > r_1$, the fixed points of $G$ are illustrated in *Appendix 2—figure 3-A to 3-D* as a function of the collective-level parameters $B$ and $T$. *Figure 7* refers to the situation where inter-colour interaction traits are smaller than intra-colour interaction traits. Here, particle populations converge in the long term to a coexistence equilibrium, where collective colour is $\phi^* = \frac{a_{11} - a_{01}}{a_{11} - a_{01} + a_{00} - a_{10}}$ (in general, different from the optimum). This equilibrium can be approached within a single collective generation if $T$ and $B$ are large (top right corner). On the other hand, when $T$ and $B$ are small (red region), the only stable fixed point invovles collectives composed solely of fast-growing particles. This corresponds to cases where individual and collective time scales (quantified by $r^{-1}$ and $T$, respectively) are insufficiently separated, or newborn size is too small, so that particle demography is essentially exponential and interactions cannot provide sufficient correction. For rapid evolution of collective colour, the most favourable starting position is one where the fixed point is closest to the optimal colour (for $\widehat{\phi} = 0.5$ this occurs for intermediate collective generation duration and bottleneck size for the trait values in *Figure 7*). Knowledge of the exact values requires, however, some preliminary measure of the ecological dynamics. Even in the absence of such information, the diagram in *Figure 7* can be used to optimise experimental design by revealing intrinsic trade-offs. A decrease in generation time, necessary for practical reasons, may for instance, be compensated by an increase in bottleneck size, without affecting the average collective phenotype.

Even when collective-level parameters are optimised so that the attractor of the $G$ function is initially close to the target colour, collective-level selection will keep acting on the particle traits, and affect phenotypic variability within the population of collectives. As stability of the fixed point increases, so to does fidelity of phenotype transmission from parent to offspring collectives. Once collective-level processes are set as to minimise the probability of collective extinction, the main

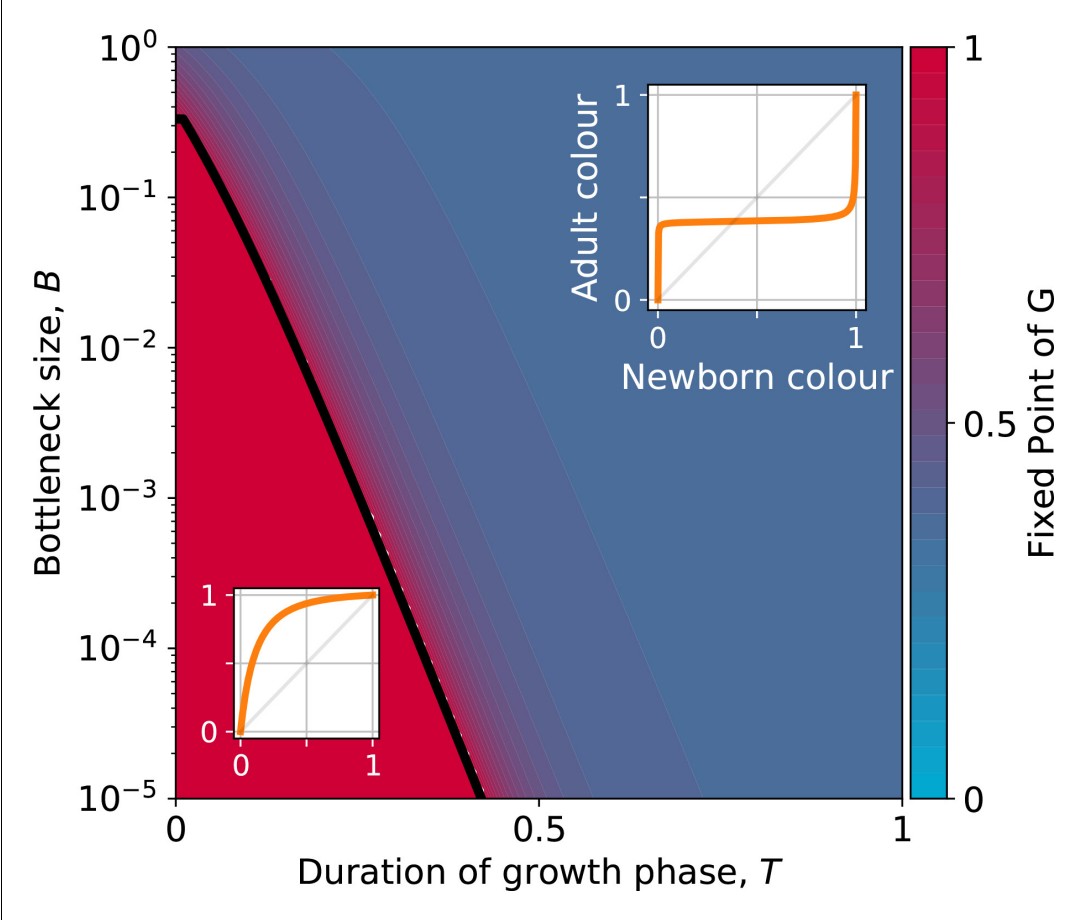

**Figure 7.** Stable fixed point of *G* as a function of collective-level parameters. Classification of the qualitative shape of the growth function and dependence on collective parameters *B* (bottleneck size) and *T* (growth phase duration). Considered here are particle traits that allow coexistence ($a_{01}<a_{11}$ and $a_{10}<a_{00}$, $r_0>r_1$, see **Appendix 2—figure 3** for the other possible parameter regions). The black line represents the limit of the region of stability of the fixed point of *G*, separating the two qualitatively different scenarios illustrated in the inset (see Appendix 2, Proposition 4 for its analytic derivation): for short collective generations and small bottleneck size, the faster growing red type competitively excludes the blue type over multiple collective generations. In order for particle types to coexist over the long term, growth rate and the initial number of particles must both be large enough for density-dependent effects to manifest at the time that selection is applied.

obstacles to evolving higher inheritance come from constrains acting on particle traits, which may limit the range of attainable *G* functions. Trade-offs on particle ecology may prevent the *G* function to attain an internal fixed point. We discuss two examples on constrained evolution in the following paragraph.

## Constrained trajectories

Thus far, we have considered evolution within a four-dimensional parameter space defined by maximum growth rates and inter-colour competition parameters. In real systems, however, constrains and trade-offs may limit the range of achievable variations in particle traits. For instance, even though faster growing particles will always experience positive selection, cell replication rate cannot increase boundlessly. Here, we consider two instances of constrained evolution, where only a subset of particle traits are allowed to mutate.

First, we consider the case where competition parameters are vanishingly small, so that *G* has no internal fixed point. Under such conditions, particle growth rates evolve to be identical (**Figure 8A**). In the absence of interactions, this is the only available solution to maintain collectives with an equal number of red and blue type particles. Under these circumstances, *G* converges to the identity function. In the deterministic approximation, collective composition remains constant in time, but

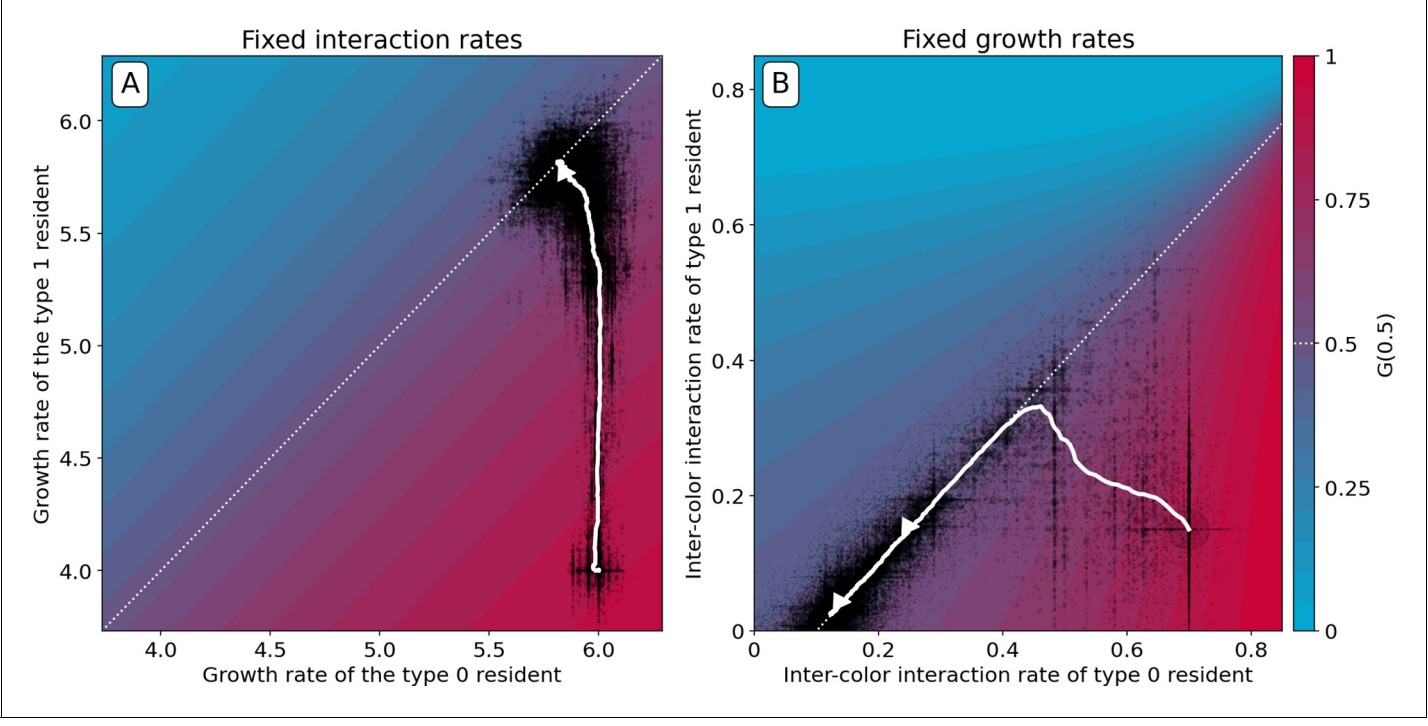

**Figure 8.** Constrained evolutionary trajectories. Dynamics through time of resident particle traits (black dots, whose size measures their abundance in the collective population) along simulated evolutionary trajectories of 300 generations, when particle-level traits are constrained. For both panels $D = 1000$, $\widehat{\phi} = 0.5$, $\rho = 20\%$, $B = 15$, and $T = 1$. The trajectory of the average resident traits is shown in white. The heatmap represents the value of $G(0.5)$ as a function of the evolvable traits, and the white dotted line indicates where collective colour is optimum. (A) Particle growth rates evolve and particles do not compete ($a^{\mathrm{inter}} = a^{\mathrm{intra}} = 0$). The evolutionary dynamics leads to alignment of growth rates ($r_0 = r_1$). (B) Inter-colour competition traits evolve and particle growth rates are constant ($r_0 = r_1 = 25$). The evolutionary dynamics first converge toward the optimality line. In a second step, asymmetric competition evolves: $a_1^{\mathrm{inter}} \to 0$ and $a_0^{\mathrm{inter}} \to a_0^{\mathrm{intra}} - a_1^{\mathrm{intra}}$. This results in a flatter $G$ function around the fixed point, providing a faster convergence to optimum colour across collective generations (*Appendix 2—figure 6*). Similar results are obtained for non-identical, but sufficiently high, growth rates.

stochastic fluctuations that cause colour to deviate from the optimum are amplified across collective generations. These deviations are nonetheless corrected in the collective population by propagating only those collectives whose colour is closest to the optimum. Such stochastic correction (*Maynard Smith and Szathmary, 1995*), however, has a high risk of failure if selection is strong and collective population size is small.

Second, we consider the case when mutations only affect the two inter-type competition parameters, while growth rates are held constant (to sufficiently high values, so that particles experience density-dependent effects in the growth phase). The evolutionary trajectory can be visualised in the plane of competition parameters $(a_{01}, a_{10})$. *Figure 8B* shows the result of a stochastic simulation superimposed to the value of $G(0.5)$. Independent of the initial values of the interaction parameters, evolution draws the system to the manifold associated with the optimal proportion $\widehat{\phi}$ (white dashed line). Evolution within this manifold is neutral in the deterministic approximation, but the presence of stochastic fluctuations drives further improvement of the fitness landscape. Correction is indeed more efficient and the distribution of collective phenotypic diversity narrower when the gradient of $G$ in the fixed point is smaller. The condition on particle traits for the latter to vanish only depends on the carrying capacities of the two particle types, and corresponds to the type with smallest carrying capacity having zero interaction rate (see Appendix 2). A similar outcome is observed when, along an evolutionary trajectory, growth rates no longer influence adult colour (*Figure 2*). Developmental correction thus selects for maximal asymmetry in interactions, whereby one particle type drives the ecological dynamics of the other type, but is itself only affected by its own type (this is fully elaborated in Appendix 2).

## Discussion

In nature, communities rarely ever qualify as units of selection in the traditional sense (*Lewontin, 1970*; *Godfrey-Smith, 2009*), because communities in nature rarely manifest heritable variance in fitness. In the laboratory, however, experimenters can exogenously impose (scaffold) Darwinian-like properties on communities such that they have no choice, but to become units of selection (*Wilson and Sober, 1989*; *Xie et al., 2019*; *Black et al., 2020*). This typically involves placement of communities in some kind of container (pot, test-tube, flask, droplet, etc.) so they are bounded and variation at the community level is thus discrete. Communities are then allowed time for individual members to replicate and interact. At the end of the 'growth' period, community function is assessed based on pre-determined criteria. The experimenter then effects replication of successful communities while discarding those that under-perform. Replication typically involves transferring a sample of individuals from a successful community to a new container replete with a fresh supply of nutrients.

Experimental and theoretical studies indicate that artificial selection on microbial communities results in rapid functional improvement (*Swenson et al., 2000a*; *Swenson et al., 2000b*; *Goodnight, 2000*; *Wade, 2016*; *Xie et al., 2019*). This is not unexpected given that experimental manipulations ensure that communities engage directly in the process of evolution by (artificial) selection as units in their own right. However, for such effects to manifest there must exist a mechanism of community-level inheritance.

Consideration of both the effectiveness of artificial selection and the problem of heredity has led to recognition that the answer likely lies in interactions (*Wilson and Sober, 1989*; *Swenson et al., 2000a*; *Swenson et al., 2000b*; *Goodnight, 2000*; *Rainey et al., 2017*). The intuition stems from the fact that in the absence of interactions, communities selected to reproduce because of their beneficial phenotype will likely fail to produce offspring communities with similar functionality. If so, then these communities will be eliminated at the next round. Consider, however, an optimal community in which interactions emerge among individuals that increase the chance that offspring communities resemble the parental type. Such an offspring community will then likely avoid extinction at the next round: selection at the level of communities is thus expected to favour the evolution of interactions because inheritance of phenotype is now the primary determinant of the success (at the community level). Indeed, simulations of multi-species assemblages have shown that evolution of interaction rates not only improves diversity-dependent fitness, but also increases collective 'heritability', defined as the capacity of randomly seeded offspring communities to reach the same dynamical state as their parents (*Ikegami and Hashimoto, 2002*; *Penn, 2003*). Further studies have stressed the role of the extracellular environment and of specific interaction networks, pointing out that microscopic constrains can affect the capacity of communities to participate in evolutionary dynamics at the higher level (*Williams and Lenton, 2007*; *Xie and Shou, 2018*; *Xie et al., 2019*).

Here, inspired by advances in millifluidics, we have developed a minimal mechanistic model containing essential ingredients of multi-scale evolution and within-community competition. We considered collectives composed of two types of particles (red and blue) that interact by density-dependent competition. By explicitly modelling demographic processes at two levels of organisation, we have obtained mechanistic understanding of how selection on collective character affects evolution of composing particle traits. Between-collective selection fuels changes in particle-level traits that feedback to affect collective phenotype. Selection at the level of communities thus drives the evolution of interactions among particles to the point where derived communities, despite stochastic effects associated with sampling at the moment of birth, give rise to offspring communities that reliably recapitulate the parental community phenotype. Such is the basis of community-level inheritance. Significantly, it has arisen from the simplest of ingredients and marks an important initial step in the endogenisation of Darwinian properties: properties externally imposed stand to become endogenous features of the evolving system (*Black et al., 2020*).

The mechanism by which particles interact to establish community phenotype is reminiscent of a development process. We have refered to this as the 'developmental corrector'. In essence, it is akin to canalisation, a central feature of development in complex living systems (*Buss, 1987*), and the basis of inheritance (*Griesemer, 2002*). Developmental correction solves the problem of implementing specific protocols for mitigating non-heritable variations in community function (*Xie et al., 2019*).

Developmental correction can be viewed as an evolutionary refinement of the stochastic corrector mechanism (*Maynard Smith and Szathmary, 1995*; *Grey et al., 1995*; *Johnston and Jones, 2015*). Both stochastic and developmental correctors solve the problem of producing enough well-formed collectives at each successive generation to prevent community-level extinction. The stochastic corrector mechanism relies on a low-fidelity reproduction process coupled to high population sizes. Deviations from successful collective states are corrected by purging collectives that depart significantly from the optimal collective phenotype. However, in the absence of strong collective-level selection the optimal community phenotype is rapidly lost. In contrast, the developmental corrector mechanism ensures that the optimal community phenotype is maintained without need for hard selection. Regardless of perturbations introduced by demography or low initial particle number, most collectives reliably reach a successful adult state. In our simulations, we show that community phenotype is maintained even in the absence of community-level selection, although ultimately mutational processes affecting particle dynamics result in eventual loss of the developmental corrector mechanism.

An operationally relevant question concerns the conditions (the initial state of the population, the nature of the scaffold and of particle-level interactions) for selection on a collective character to result in evolution of developmental correction. While we did not detail the probability of collective lineage extinction, it is possible that collectives become monochromatic before evolution has had time to act on particle traits. In such cases, which are more likely if particle-level traits are far from the region of coexistence, and if time-scales of particle and collective generations are not well separated, then collective-level evolution will grind to a halt. In all other cases, provided there are no other evolutionary constraints, selection will eventually lead the system toward regions of particle trait-space where the collective phenotype becomes reliably re-established. The efficiency of this selective process and its transient behaviour depend on collective-level parameters that control growth and reproduction.

From our individual-based simulations and ensuing deterministic approximation, it is clear that once density-dependent interactions govern the adult state, then collective-level selection for colour is promptly effected. This happens provided the intra-collective ecology lasts long enough for non-linear effects to curb particle growth. When this is not the case, for example when the bottleneck at birth is small, or collective-level generation time is too short, evolution of developmental correction will be impeded. The latter favours rapidly growing particles (*Abreu et al., 2019*) and offers little possibility for the evolution of developmental correction. When the ecological attractor within collectives leads to the extinction of one of the two types, long collective-level generation times are incompatible with the maintenance of diversity (*van Vliet and Doebeli, 2019*). However, in our model, particle-level evolution changes the nature of the attractor from extinction of one of two types to stable coexistence, and concomitantly particle and collective time-scales become separated. Even before developmental correction becomes established, evolution can transiently rely on stochastic correction to ensure the maintenance of particle co-existence.

There are two aspects to heredity: resemblance — the extent to which reproduction and development maintain the average offspring phenotype — and fidelity (or determination) — a measure of phenotypic variance (*Jacquard, 1983*; *Bourrat, 2017*). In our model, resemblance is established once density-dependent interactions counter the bias toward fast replicating particles: when the $G$ function has an internal fixed point in $\hat{\phi}$, systematic drift of average collective colour is prevented. The increase in resemblance is associated with progressive divergence of particle and collective demographic time scales. As a consequence, the collective phenotype is placed under the control of particle traits rather than demographic stochasticity. On a longer time scale, fidelity improves by subsequent changes in interaction parameters under the constraint that they do not affect average adult colour. The variance of the phenotype around the optimum is reduced by increasing canalisation (flattening of the $G$ function). This is best achieved by a strong asymmetry in the competition traits, whereby one type has a logistic, uncoupled, dynamic, and the second type adjusts its growth to the former's density. Interestingly, it is always the type with the lower carrying capacity, regardless of its relative growth rate, that acts as the driver.

The relationship between parameters on very long time scales, when the adult colour is essentially dependent on interaction rates, depends critically on the space of possible values particle traits can assume. For instance, our analysis took into account only competitive intractions between colours.

Extension of the deterministic approximation to cases when ecological interactions between colours are exploitative or mutualistic indicates that selection for collective coulour can drive changes in the very nature of the interactions so as to make them progress towards less and less reciprocally harmful coexistence (for full elaboration see Appendix 3).

Our goal has been to produce a simple, tractable scenario for studying the effects of artificial selection on collectives, which while theoretical, is firmly connected to plausible biological experiments. The model could be extended in multiple ways in order to analyse the effects of additional factors, including impact of non-overlapping generations and variation in the timing of reproduction (which would introduce an element of bet-hedging), of migration and mixing between collectives (which could be akin to gamete production and zygote formation), and inclusion of more than two kinds of particle types. More complex selective regimes can also be envisaged, such as those that reward collectives based on absolute population size of particle types, which would allow less abstract collective functions to be considered. However, regardless of these refinements, we suspect that our core conclusion will stand firm: collective-level selection favours particle dynamics that improve collective-level heredity. The ability to reliably re-establish successful adult states of past-generations from simpler and potentially noisy initial conditions is adaptive at the collective level.

The mechanism of developmental correction is broadly relevant and extends beyond cells and communities to particles of any kind that happen to be nested within higher-level self-replicating structures. As such, the mechanism of developmental correction may be relevant to the early stages in each of the major (egalitarian) evolutionary transitions in individuality (*Queller, 1997*; *Maynard Smith and Szathmary, 1995*), where maintenance of particle types in optimal proportions was likely an essential requirement. For example, it is hard to see how protocells cells evolved from lower level components (*Takeuchi and Hogeweg, 2009*; *Baum and Vetsigian, 2017*), chromosomes from genes (*Smith and Szathmáry, 1993*), and the eukaryotic cell from independent bacterial entities (*Martin and Müller, 1998*) without some kind of self-correcting mechanism acting at the collective level.

Beyond these fundamental considerations, the mechanism of developmental correction and the ecological factors underpinning its evolution have important implications for top-down engineering of microbial communities for discovery of new chemistries, new functions, and even new organisms. The minimal recipe involves partitioning communities into discrete packages, provision of a period of time for cell growth, selective criteria that lead to purging of sub-optimal collectives and reproduction of optimal collectives to establish the next generation of collectives. These manipulations are readily achieved using millifluidic devices that can be engineered to operate in a Turing-like manner allowing artificial selection on community-level traits across thousands of independent communities. As mentioned above, critical tuneable parameters beyond number of communities, mode of selection and population size, are duration of collective generation time and bottleneck size at the moment of birth.

The extent to which the conclusions based on our simple abstract model are generally applicable to the evolution of more complex associations, such as symbioses leading to new forms of life, will require future exploration of a broader range of particle-level ecologies. Possibilities to make community dynamics more realistic by complexifying mathematical descriptions of particle-level processes are plentiful (*Williams and Lenton, 2007*; *Zomorrodi and Segrè, 2016*). Of particular interest for the evolution of efficient developmental correction are cases when community ecology has multiple attractors (*Penn and Harvey, 2004*), is highly sensitive to initial conditions (*Swenson et al., 2000b*), or presents finite-effect mutations sustaining 'eco-evolutionary tunnelling' (*Kotil and Vetsigian, 2018*). Besides enlarging the spectrum of possible within-collective interactions, future relevant extensions might explore the role of physical coupling among particles and of horizontal transmission between collectives (*van Vliet and Doebeli, 2019*) in enhancing or hampering efficient inheritance of collective-level characters.

## Methods

Methods are described in Appendix 1, 2 and 3, interspersed with more technical descriptions of the results presented in the main text.

## Acknowledgements

We thank Wenying Shou, Sara Mitri and Alvaro Sanchez for review of the manuscript and valuable comments. We also thank Benjamin Kerr, Will Ratcliff, Eric Libby, Michael Doebeli, Pierrick Bourrat and Philippe Nghe as well as the members of our respective teams for their feedback. GD was funded by the *Origines et Conditions d'Apparition de la Vie (OCAV)* programme, PSL University (ANR-10-IDEX-001–02), awarded to PBR, SDM and AL; SDM was supported by the French Government under the program Investissements d'Avenir (ANR-10-LABX-54 MEMOLIFE and ANR-11-IDEX-0001-02PSL), AL thanks the Center for Interdisciplinary Research in Biology (CIRB) for funding.

## Additional information

### Competing interests

Paul B Rainey: Reviewing editor, *eLife* and founding member of MilliDrop. The other authors declare that no competing interests exist.

### Funding

| Funder | Grant reference number | Author |
| --- | --- | --- |
| Agence Nationale de la Recherche | ANR-10-IDEX-001-02 | Guilhem Doulcier<br>Paul B Rainey |
| Agence Nationale de la Recherche | ANR-10-LABX-54 | Silvia De Monte |
| Agence Nationale de la Recherche | ANR-11-IDEX-0001-02 | Silvia De Monte |
| Max-Planck-Gesellschaft | | Paul B Rainey |

The funders had no role in study design, data collection and interpretation, or the decision to submit the work for publication.

### Author contributions

Guilhem Doulcier, Software, Formal analysis, Visualization, Methodology; Amaury Lambert, Formal analysis, Supervision, Methodology; Silvia De Monte, Resources, Formal analysis, Supervision, Methodology; Paul B Rainey, Conceptualization, Resources, Supervision, Funding acquisition

### Author ORCIDs

Guilhem Doulcier https://orcid.org/0000-0003-3720-9089
Amaury Lambert http://orcid.org/0000-0002-7248-9955
Silvia De Monte https://orcid.org/0000-0001-7953-5494
Paul B Rainey https://orcid.org/0000-0003-0879-5795

### Decision letter and Author response

Decision letter https://doi.org/10.7554/eLife.53433.sa1
Author response https://doi.org/10.7554/eLife.53433.sa2

## Additional files

### Supplementary files

• Source code 1. Source code for simulation and figure generation. Also available at https://gitlab.com/ecoevomath/estaudel.

• Transparent reporting form

## Data availability

The source code for all simulations and figures in the manuscript is available as a zip file uploaded with the manuscript and in a public git repository (https://gitlab.com/ecoevomath/estaudel; copy archived at https://github.com/elifesciences-publications/estaudel).

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

# Appendix 1

## Stochastic model

This appendix presents an outline for the approximated stochastic simulation of nested population dynamics described in the main text. We first attend to the case where particle populations are monomorphic without mutation and then move to consider the role of particle-level mutation. Full implementation of the model is available as *Source code 1* and at https://gitlab.com/ecoevomath/estaudel (*Doulcier, 2020*).

### Parameters

First, we list the parameters of the numerical model introduced in the main text. The collective selection regime is specified by a set of collective-level parameters that are kept constant along the evolutionary trajectory.

```
// Parameters
D ← 1000                    // Number of collectives
M ← 10,000                  // Number of collective generations
B ← 15                      // Bottleneck size
T ← 1.0                     // Duration of a collective generation
regime ← selective          // Selective or Neutral regime
ρ ← 0.2                     // Fraction of extinguished collectives
φ̂ ← 0.5                     // Optimal collective colour (for selective regime)
```

Collectives are comprised of two kinds of self-replicating particle (red and blue) that carry a different set of traits. Traits can mutate (see the mutation section below), and are represented as global variables.

```
// Particle traits
// Carried by red particles
r₀              // Maximum growth rate
a₀ᶦⁿᵗʳᵃ          // Competition with red particles
a₀ᶦⁿᵗᵉʳ          // Competition with blue particles
//Carried by blue particles
r₁              // Maximum growth rate
a₁ᶦⁿᵗʳᵃ          // Competition with blue particles
a₁ᶦⁿᵗᵉʳ          // Competition with red particles
```

Particle traits:
- Carried by red particles:
  - $r_0$ // Maximum growth rate
  - $a_0^{\text{intra}}$ // Competition with red particles
  - $a_0^{\text{inter}}$ // Competition with blue particles
- Carried by blue particles:
  - $r_1$ // Maximum growth rate
  - $a_1^{\text{intra}}$ // Competition with blue particles
  - $a_1^{\text{inter}}$ // Competition with red particles

The state variables ($D \times M$ matrices) store the adult state of collectives along a trajectory.

```
//State variables
N₀          //number of blue individuals in each collective at each generation
N₁          //number of red individuals in each collective at each generation
Φ           //proportion of red individuals in each collective at each generation
```

### Initial conditions

Initial conditions consist in defining the number of red and blue particles in each collective at the beginning of generation 0.

```
procedure INITIAL CONDITIONS
    x₀ ← 0.5     //Initial red-blue ratio
    for d from 1 to D do
        N₀[d,0] ← RandomBinomial(B,x₀)
        N₁[d,0] ← B − N₀[d,0]
```

### Outline of the main loop

The main loop of the algorithm applies the sequence growth-selection-reproduction for each generation.

```
procedure MAIN LOOP
  for m from 1 to M do
    //Particle Growth
    for d from 1 to D do
      N₀[d,m], N₁[d,m] ← GROWTH(N₀[d,m], N₁[d,m], r₀, r₁, a₀ᶦⁿᵗʳᵃ a₀ᶦⁿᵗᵉʳ, a₁ᶦⁿᵗʳᵃ a₁ᶦⁿᵗᵉʳ)
      Φ[d,m] ← N₁[d,m]/(N₀[d,m] + N₁[d,m])
    // Collective − level selection
    parents ← SELECT_COLLECTIVES(Φ[d,m], ρ, φ̂)
    // Collective − level reproduction
    for d from 1 to D do
      N₀[d,m+1] ← RandomBinomial(B, Φ[parents[d],m])
      N₁[d,m+1] ← B − N₀[d,m+1]
```

## Particle-level ecology

The ecological dynamics of particles is expressed by a multi-type birth-death process with a linear birth rate and a linearly density-dependent death rate. Each type of particle $i$ is characterised by four traits: colour ($c_i$, red or blue), maximum growth rate $r_i$, and two density-dependent interaction parameters. Particles of the same colour interact according to $a_i^{\mathrm{intra}}$, whereas particles of different colour interact according to $a_i^{\mathrm{inter}}$. Interaction terms are in the order of 0.1 and scaled by a carrying capacity term $K$. The dynamic is modelled by a continuous-time Markov jump process with rates:

| Each particle of type i . . . | With rate. . . |
| --- | --- |
| Reproduces (add a particle of type i) | $r_i$ |
| Dies (remove a particle of type i) | $r_i(\sum_j \delta_{c_i=c_j} x_j a_j^{\mathrm{intra}} K^{-1} + \sum_j \delta_{c_i \neq c_j} x_j a_j^{\mathrm{inter}} K^{-1})$ |

$\delta_{i=i} = 1$ if $i = j$ or 0 if $i \neq j$. Additionally $\delta_{i \neq i} = 1$ if $i \neq j$ or 0 if $i = j$.

The stochastic trajectory of the system is simulated using a Poissonian approximation used in the basic $\tau-$leap algorithm (**Gillespie, 2001**), $dt$ is chosen to be small enough so that population size never becomes negative.

```
function GROWTH (n₀, n₁, r₀, r₁, a₀ᶦⁿᵗʳᵃ a₀ᶦⁿᵗᵉʳ, a₁ᶦⁿᵗʳᵃ a₁ᶦⁿᵗᵉʳ)
  //Stochastic simulation of the population dynamics
  while t do
    birth0 ← RandomPoisson(dt × n₀r₀)
    birth1 ← RandomPoisson(dt × n₁r₁)
    death0 ← RandomPoisson(dt × n₀r₀(n₀a₀ᶦⁿᵗʳᵃ + n₁a₁ᶦⁿᵗᵉʳ))
    death1 ← RandomPoisson(dt × n₁r₁(n₀a₀ᶦⁿᵗᵉʳ + n1a₁ᶦⁿᵗʳᵃ))
    n₁ ← n₁ + birth1 − death1
    n₀ ← n₀ + birth0 − death0
    t ← t + dt
  Return n₀, n₁
```

Early tests using an exact stochastic simulation algorithm (Doob-Gillespie SSA, **Gillespie, 1976**) did not exhibit qualitative changes in the trajectory, but greatly increased the computation duration.

## Collective-level selection

The collective-level selection phase consists in associating each of the $D$ new collectives from generation $m + 1$ with a single parent at generation $m$. In the main text we contrast two regimes: colour-neutral and colour-selective. In both cases, a fixed proportion $\rho$ of the collective population at generation $m$ is marked for extinction. In the neutral regime, collectives to be eliminated are selected uniformly at random (**Appendix 1—figure 1A**), whereas in the selective regime they are those that rank the highest in their distance to the optimal colour $\hat{\phi}$ (**Appendix 1—figure 1B**). Each surviving collective produces offspring. Moreover, the remaining collectives from generation $m + 1$ are generated by a parent chosen uniformly at random from the set of surviving collectives.

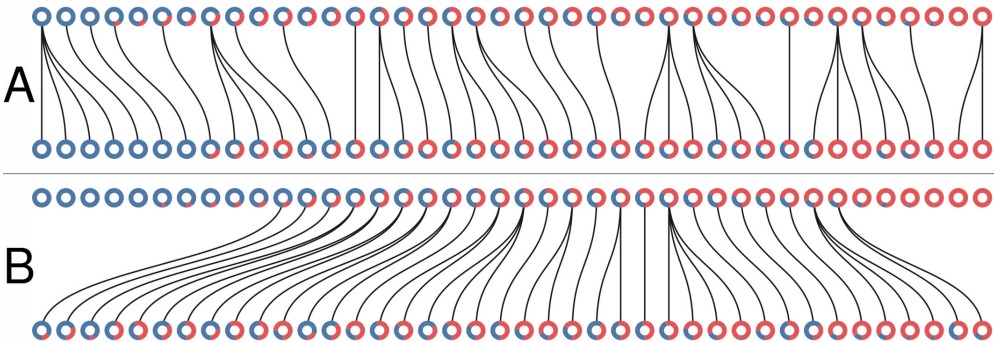

**Appendix 1—figure 1.** Collective-level selection regimes. Each ring represents a collective. The blue section of the ring represents the proportion of blue particles at the adult stage of the collective. Parent and offspring are linked by a black line. (**A**) One generation of the neutral regime. (**B**) One generation of the selective regime. In both cases $M = 1$, $D = 40$, $\rho = 0.4$.

```
function SELECT.COLLECTIVES (φ, ρ, φ̂, regime)
    //Return the indices of the new collectives' parents
    surviving ← empty list// indices of the surviving collectives
    reproducing ← empty list// indiceof the reproducing collectives
    parents ← empty list// indices of the parent collectives
    if regime is selective then
        threshold ← Percentile((φ − φ̂)², ρ)
        for d from 1 to D do
            if (φ[d] − φ̂)² < threshold then
                Add d to surviving
    else if regime is neutral then
        surviving = RandomMultinomialWithoutReplacement(1...d, n = (1 − ρ)D)
    //Extinct collectives are replaced by the offspring of a randomly drawn surviving collective
    for 1 to D-Length(surviving) do
        Add RandomChoice(surviving) to reproducing
    //Surviving collectives have at least one offspring, and the population
    size is kept constant by additional reproduction events.
    parents ← Concatenate(surviving, reproducing)
    Return parents
```

Other selection procedures can be implemented, such as randomly sampling $\rho D$ collectives with weight based on the colour-deviation to the optimal colour. A non-exhaustive exploration of other selection rules indicates that the qualitative results of the model are robust to changes in the selective regime, as long as collectives with an optimal colour are favoured and the collective population does not go extinct.

Collective reproduction is implemented by seeding an offspring collective with a sample of $B$ particles drawn according to the proportion of the parent collective. We assume that the final particle population sizes are big enough so that each reproduction event can be modelled as an independent binomial sample. Smaller population sizes might require simultaneous multinomial sampling of all offspring.

## Mutation of particle traits

The complete model adds the possibility for particle trait ($r, a^{inter}$) to mutate. Each collective contains two variants of each colour. Whenever a variant goes extinct, the remaining type is called the 'resident', and a mutant type is created as follows. First, traits of the resident are copied in one newborn particle, then one of the mutable traits — either the growth rate ($r$) or the inter-colour competition trait ($a_{inter}$) — is chosen at random, and finally a random value is added that is taken from a uniform distribution over $[-\varepsilon, \varepsilon]$ (in **Figure 2**, $\varepsilon = 0.1$). Traits are kept positive by taking the absolute value of the result. This process is in the spirit of adaptive dynamics in which invasion of a single new mutant is repeatedly assessed in a monomorphic population.

We checked that relaxing this assumption (i.e. allowing more than two types of each colour in each collective), or waiting for rare mutations to appear did not change the qualitative results. The pseudo-code outlined above was modified in order to track the trait value of both resident and mutant types in each collective, rather than having the trait values as global variables.

## Variability of the derived particle traits

The single trajectory of the stochastic model discussed in the main text (*Figure 2*) is obtained by setting initial conditions for both collective composition (fraction of red and blue particles in each collective at the beginning of the first collective generation) and for the value of particle-level parameters that are subjected to mutations. We used this specific trajectory to illustrate key features of the evolutionary outcome, that are then explained by the deterministic approximation (Appendix 2). Such approximation, discussed below, allows exhaustive exploration of parameter space, that would otherwise prove impossible because of computation demands. However, it provides only indirect insight on the expected variance of the corresponding particle-based stochastic process. Here, we address variability of the evolutionary outcome of stochastic simulations in a few relevant cases.

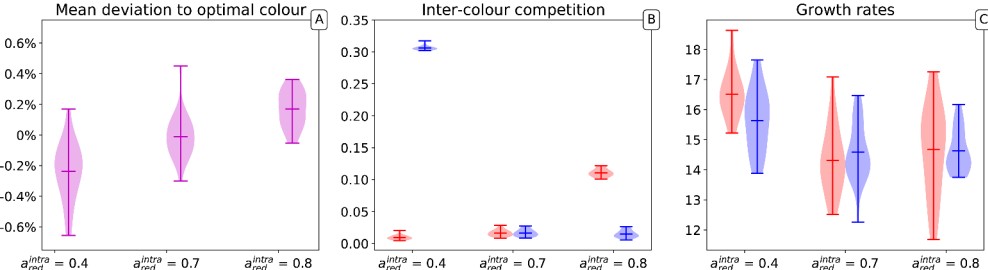

**Appendix 1—figure 2.** Distribution of average mutable traits values across several simulations. Three parameters sets are represented in which red and blue types have respectively equal carrying capacity ($a_0^{\mathrm{intra}} = a_1^{\mathrm{intra}} = 0.7/K$), red types have a higher carrying capacity ($a_0^{\mathrm{intra}} = 0.4$) and lower carrying capacity ($a_0^{\mathrm{intra}} = 0.8$). The results of 20 independent simulations for each of the parameters sets are shown. For each simulation, $D = 100$ collectives, $M = 10,000$ generations, $B = 15$ particles, the optimal collective colour is $\widehat{\phi} = 0.5$ and initial particle traits are $(r_1 = r_0 = 5, a_1^{\mathrm{intra}} = 0.7/K, a_0^{\mathrm{inter}} = a_1^{\mathrm{inter}} = 0.3/K)$, with $K = 1500$.

*Appendix 1—figure 2* illustrates the distribution of average colour in a population of collectives (A) and of the average associated traits (interaction rates [B]; growth rates [C]) after $M = 10,000$ collective generations for three different values of the red type's carrying capacity. For each set of parameters, 20 trajectories where simulated. In none of the 60 simulations do populations became entirely monochromatic.

The evolutionary outcome of the stochastic model is highly reproducible, with only a small percentage variation in the average collective colour achieved in different realizations of the stochastic process. Apart from a small systematic bias toward the type with the higher carrying capacity (i.e. the lower $a^{\mathrm{intra}}$), changing the carrying capacity does not affect the ability of selection to achieve the target colour (*Appendix 1—figure 2A*).

The effects of collective selection on particle traits instead change depending on the role of such traits in determining the collective colour. *Figure 2* of the main text illustrates the convergence, over one single evolutionary trajectory, of the inter-colour interaction parameters to specific values. Correspondingly, the average of these parameters in the population also changes little across multiple realizations of the stochastic evolutionary process (*Appendix 1—figure 2B*). As expected, the values depend on the parameters that do not evolve, that in the cases illustrated here, are the carrying capacities. However, one can notice that as long as interaction rates are bounded to be positive, one interaction (in the specific case where carrying capacities are equal) will in the long run vanish, indicating that one of the types drives the intra-collective dynamics. This corresponds to predictions, as discussed in Appendix 2 and 3.

Maximal growth rates, that were observed to change within one single realization of the evolutionary trajectory, also vary considerably across realizations (*Appendix 1—figure 2C*). This is a

consequence of their quasi-neutrality with respect to collective selection: once the particle population approaches a steady-state within any collective generation, the speed at which it grows at low density is less important.

## Different target colours

In the main text, we have considered that the target of selection was a situation when collectives are composed of half red and half blue particles. Here, we repeated the numerical simulations for different target proportions of the two types and discuss how the choice of $\hat{\phi}$ affects the outcome of the evolutionary trajectory. We performed ten independent simulations for 10,000 collective generations, starting from the same initial conditions and parameter values, but varying the target of selection.

*Appendix 1—figure 3A* shows that different target colours, that is, compositions of the collectives, can be selected with an overall accuracy of a few percent. However, the variability among simulations is larger when the target is strongly skewed toward one or another monochromatic state. This reflects an increased variability in the selected particle traits, which manifests when the derived interaction rates are considered (*Appendix 1—figure 3B*) and maximal growth rate (*Appendix 1—figure 3B*). While maintaining the previously observed lower variability in interaction rates than in growth rates, all derived traits change more from run to run when the target is extreme. This is likely the consequence of a lower effectiveness in the action of collective-level selection, as a considerable part of the population ends up being monochromatic just because of sampling at birth. As a consequence, convergence towards the target is also considerably slower.

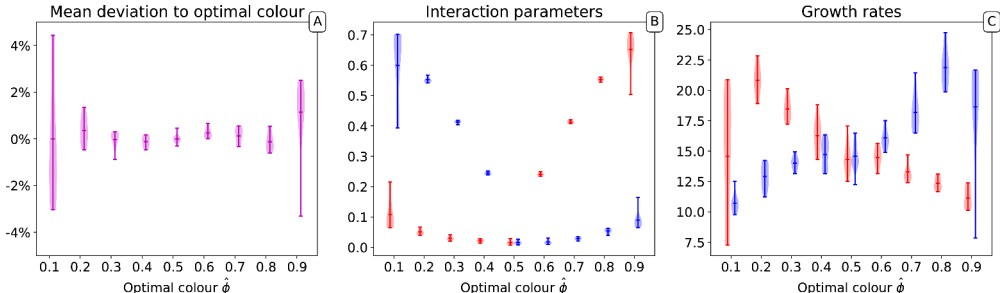

**Appendix 1—figure 3.** Distribution of the final traits across several experiments for different optimal colours. Ten independent simulations are performed for each value of $\hat{\phi}$. In all simulations, $D = 100$ collectives, $M = 10,000$ generations, $B = 15$ particles, and initial particle traits are $(r_0 = r_1 = 5, a_0^{\text{intra}} = a_1^{\text{intra}} = 0.7/K, a_0^{\text{inter}} = a_1^{\text{inter}} = 0.3/K)$, with $K = 1500$.

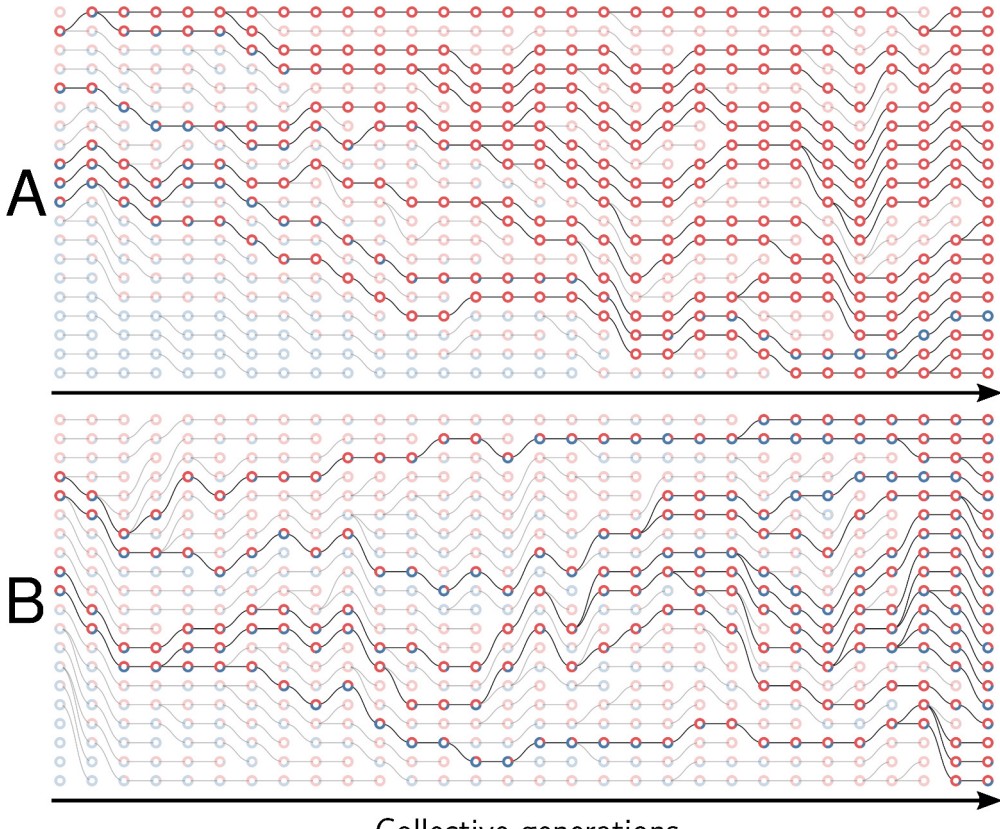

**Appendix 1—figure 4.** Example of collective genealogy (Supplement of *Figure 2*). Symbols and colours are as in *Appendix 1—figure 1* and extinct lineages are marked transparent. Collective-level parameters in this simulation are $M = 30$, $D = 20$, $\rho = 0.1$. *A*. Neutral regime: at the final generation, collectives are monochromatic and most likely composed of the faster-growing type. *B*. Selective regime: at the final generation, collectives contain both red and blue particles.

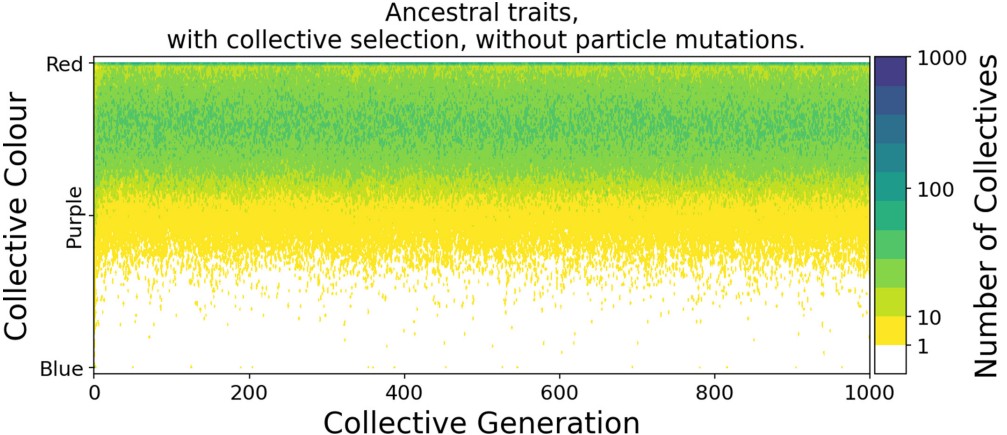

**Appendix 1—figure 5.** The stochastic-corrector mechanism can maintain both types of particles in the absence of mutations (Supplement of *Figure 2*). Collective phenotype distribution through time in the selective regime with ancestral particle traits and no mutations. Without collective-level mutation, the only mechanism maintaining both types within the population is the stochastic corrector, whereby a fraction of the collectives with colour closer to the target are propagated to the next collective generation. This means that at every generation the distribution of collective phenotypes is skewed towards the colour that has higher maximal growth rate, and the target colour is realized, in a small fraction of collective population, thanks to stochastic fluctuations in the

composition at birth. Parameters are $D = 1000$, $M = 1000$, $\widehat{\phi} = 0.5$, $B = 15$ and initial traits ($r_0 = 6$, $a_0^{\text{intra}} = 0.8/K$, $a_0^{\text{inter}} = 0.15/K$, $c_0 = \text{red}$) for red particles and ($r_1 = 4$, $a_1^{\text{intra}} = 0.3/K$, $a_1^{\text{inter}} = 0.15/K$, $c_1 = \text{blue}$) for blue particles, with $K = 1500$. The initial proportions at generation 0 were randomly drawn from a uniform distribution.

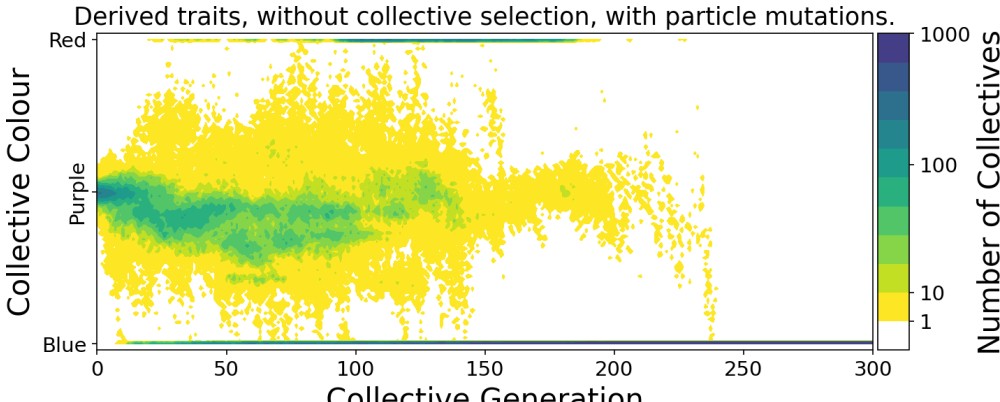

**Appendix 1—figure 6.** Particle trait mutations lead to slow loss of optimal collective colour after removal of collective-level selection (Supplement of *Figure 4*). Modification of the collective phenotype distribution when particle traits mutate and no selection for colour is applied, starting from the particle traits after 10,000 generations of selection for purple colour (as in *Figure 4B*). Collectives continue to produce purple offspring for more than 200 generations, before drift of particle traits erodes developmental correction. In contrast, for the ancestral particle traits, lineages become monochromatic in less than 10 generations (*Figure 4A*).

## Appendix 2

### Lotka-Volterra deterministic particle ecology

## Model for intra-collective dynamics

The stochastic ecological dynamics of red ($N_0$) and blue ($N_1$) particles within a collective simulated by the algorithm described in Appendix 1 is approximated (see *Figure 3*) by the deterministic competitive Lotka-Volterra Ordinary Differential Equation:

$$\begin{cases} \frac{dN_0}{dt} = r_0 N_0 \left(1 - \frac{a_{00}}{K}N_0 - \frac{a_{01}}{K}N_1\right) \\ \frac{dN_1}{dt} = r_1 N_1 \left(1 - \frac{a_{10}}{K}N_0 - \frac{a_{11}}{K}N_1\right) \end{cases} \tag{1}$$

Here, $\mathbf{r} = (r_0, r_1)$ is the pair of maximal growth rates for red and blue particles. Since the system is symmetric, we consider only the case where red particles grow faster than blue particles ($r_0 > r_1$). The effect of pairwise competitive interactions between cells are encoded in the matrix $\mathbf{A} = (a_{ij})_{i,j \in \{0,1\}^2}$. All competitive interactions are considered harmful or neutral ($0 \leq a_{ij}$) (see Appendix 3 for more general types of interactions). Here and in the main text, we consider that the four free parameters can evolve at the same time. In the last section of the Results in the main text we explore cases when their variation is constrained by trade-offs.

So as to explore the space of all possible interaction intensities, no specific mechanism of interaction is assumed, and thus qualitatively different ecological dynamics of the particle populations. Evolutionary trajectories constrained by particle traits are briefly discussed in the last section of the Results (main text).

The carrying capacity of a monochromatic collective is $\frac{K}{a_{00}}$ for red particles and $\frac{K}{a_{11}}$ for blue particles. $K$ is a scaling factor for the intensity of pairwise interactions that can be used to rescale the deterministic system to match the stochastic trajectories. Without loss of generality, we thus assume that $K = 1$.

A natural set of alternate coordinates for the system in *Equation 1* are total population size $N := N_0 + N_1$ and collective colour, defined as the frequency of red individuals $x := \frac{N_0}{N}$. In these coordinates, the deterministic dynamics are the solution to the following ODE.

$$\begin{cases} \frac{dN}{dt} = Ng(x,N) \\ \frac{dx}{dt} = x(1-x)h(x,N) \end{cases} \tag{2}$$

The functions $g$ and $h$ are polynomials in $x$ and $N$, of coefficients:

| Monomial | Coefficient in $h$ | Monomial | Coefficient in $g$ |
|---|---|---|---|
| 1 | $r_0 - r_1$ | 1 | $r_1$ |
| $N$ | $a_{11}r_1 - a_{01}r_0$ | $N$ | $-a_{11}r_1$ |
| $Nx$ | $r_1(a_{10} - a_{11}) + r_0(a_{01} - a_{00})$ | $x$ | $r_0 - r_1$ |
| | | $xN$ | $r_1(2a_{11} - a_{10}) - r_0 a_{01}$ |
| | | $x^2N$ | $r_0(a_{01} - a_{00}) + r_1(a_{10} - a_{11})$ |

*Appendix 2—figure 1* shows an example of the ODE flow in both coordinate systems. Two unstable trivial equilibria and a stable coexistence equilibrium are located at the intersection of the isoclines. Within one collective generation, the dynamics follow such flows for duration $T$, starting from initial conditions on the line $N_0 = B - N_1$ (that is, $N = B$).

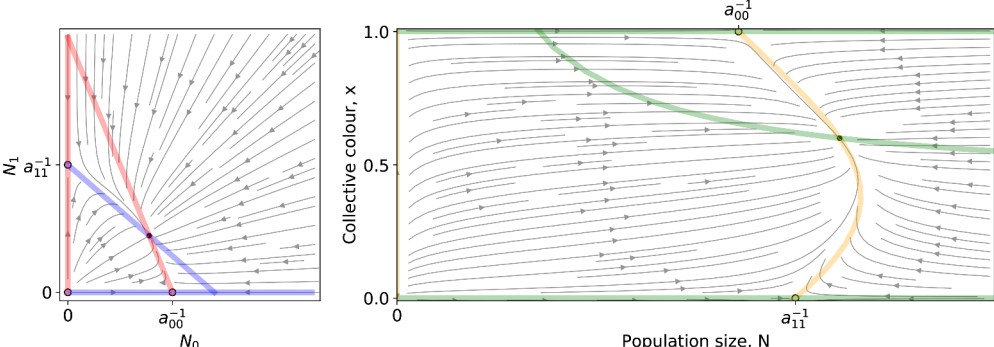

**Appendix 2—figure 1.** Left: A Lotka-Volterra flow (*Equation 1*) in $(N_0, N_1)$ coordinates. Right: The same flow in $(N, x)$ coordinates. Coloured lines are the null isoclines. Empty (resp. filled) circles mark unstable (resp. stable) equilibria.

Even though *Equation 1* is more directly related to the individual-based stochastic simulation, the dynamics of collective colour are understood more easily using *Equation 2*. Therefore, in the following we will use the latter formulation.

The dynamics of particle types across collective generations are modelled as a piecewise continuous time change (*Appendix 2—figure 2*), where $x^m(t)$ is the fraction of red particles at time $t \in [0, T]$ during collective generation $m$.

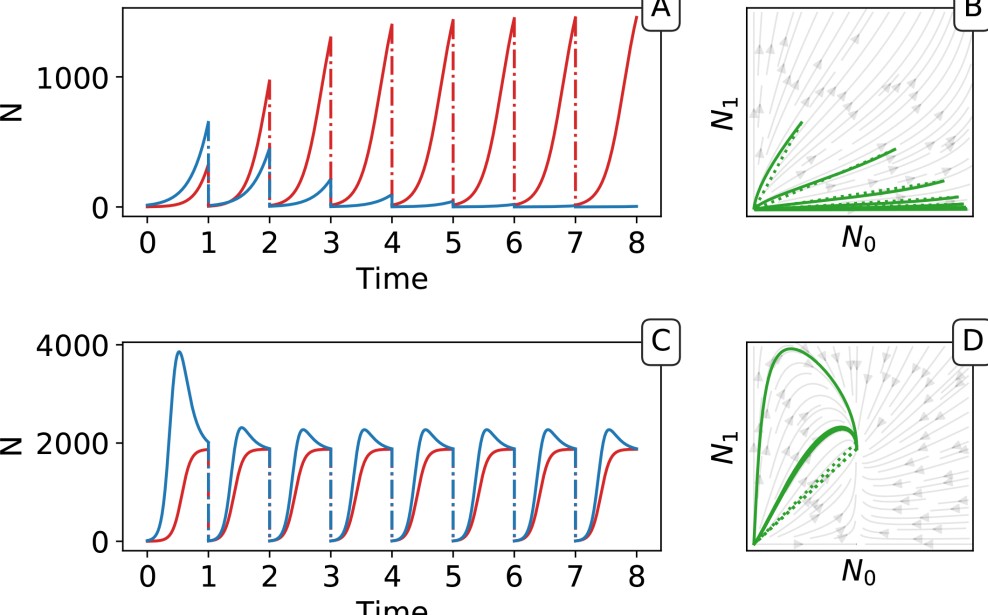

**Appendix 2—figure 2. Piecewise continuous trajectory.**
Iterating the deterministic model yields a piecewise continuous trajectory in the $(N_0, N_1)$ space. The growth phase (continuous lines) alternates with the dilution (dotted lines). In (**A, B**) traits are taken from generation 3 and in (**C,D**) from generation 9000 of the main simulation with selection (*Figure 2*). Successive adult states can be computed using the $G_\theta$ function as a recurrence map (see *Figure 5* in the main text).

We focus in particular on the succession $N^m(T), x^m(T)$ of collective ''adult'' states at the end of each successive generation $m$. In the following we note $\phi^m = x^m(T)$, which is the adult colour of the collective at the end of the growth phase. At the beginning of each collective generation, we impose that, regardless of the number of cells in the parent, every collective contains the same number of cells $N^m(0) = B \in \mathbb{R} \forall m$. In contrast, the newborn collective colour $x^m(0)$ depends on the colour of the

parent. In this deterministic model, we consider that there is no stochastic variation or bias due to sampling at birth, so the collective colour of the newborn is equal to that of its parent: $x^m(0) = x^{m-1}(T) = \phi^{m-1}$. In the first generation ($m = 0$), the population size is $B$ in every collective and the fraction of red particles is chosen uniformly at random between 0 and 1.

The proportion of red particles at adult stage $\phi^m$ is obtained from these initial conditions $(B, \phi^{m-1})$ by integrating *Equation 2*. Since these equations are not explicitly solvable, there is no analytic expression for the result of the transient dynamics. However, having constrained the initial conditions to a single dimension, the adult colour is a single-valued function of the initial composition of the collective, defined as follows.

## Definition (Growth function)

Given the set of particle traits $\theta := (\mathbf{r}, \mathbf{A}) \in E := [0, \infty)^2 \times [0, \infty)^4$, the bottleneck size $B \in (0, \infty)$ and the duration of the growth phase $T \in (0, \infty)$, the growth function $G_\theta$ is defined as the application that maps an initial proportion of red particles $\phi$ to the proportion of red particles after duration $T$. Thus, $G_\theta(\phi, T, B) = x(T)$, with $x(t)$ is the unique solution to the following Cauchy problem:

$$\begin{cases} \frac{dN}{dt} &= Ng(x,N) \\ \frac{dx}{dt} &= x(1-x)h(x,N) \\ N(0) &= B \\ x(0) &= \phi \end{cases} \tag{3}$$

To simplify notations in the main text, we set $G(\phi) = G_\theta(\phi, B, T)$.

## Relation with the stochastic evolutionary model

As explained in the main text, the growth function $G_\theta$ defines the recurrence relation between the colour of the newborn offspring and its colour at adulthood. If the reproduction process entails no stochasticity, the latter also defines the composition of collectives at the next generation. As a consequence, the iterative application of $G_\theta$ approximates the change in time of the adult colour when a population with a given, fixed, set of parameters and traits is transferred across collective generations.

The fixed points of this discrete-time system allow understanding and classifcation of behaviours observed along stochastic evolutionary trajectories. When mutation rate of particle traits is sufficiently small and growth rates are not vanishing, collectives approach such fixed points in a few collective generations by iteration of the $G_\theta$ function (*Figure 5*). The evolutionary trajectory can thus be seen as a succession of fixed points of $G_\theta$. The surface of the fixed points of $G_\theta$ as a function of the particle parameters can be computed numerically, as well as its dependence on the collective generation duration $T$ and bottleneck size $B$. A small number of qualitatively different configurations are possible for the fixed points (illustrated in *Appendix 2—figure 3 1-4*). In particular, the case in which $G_\theta$ possesses an internal, stable fixed point (*Appendix 2—figure 3*) constitutes the optimal solution to constant selection for collective colour, in that it ensures the highest degree of colour reproducibility, on average, across collective generations.

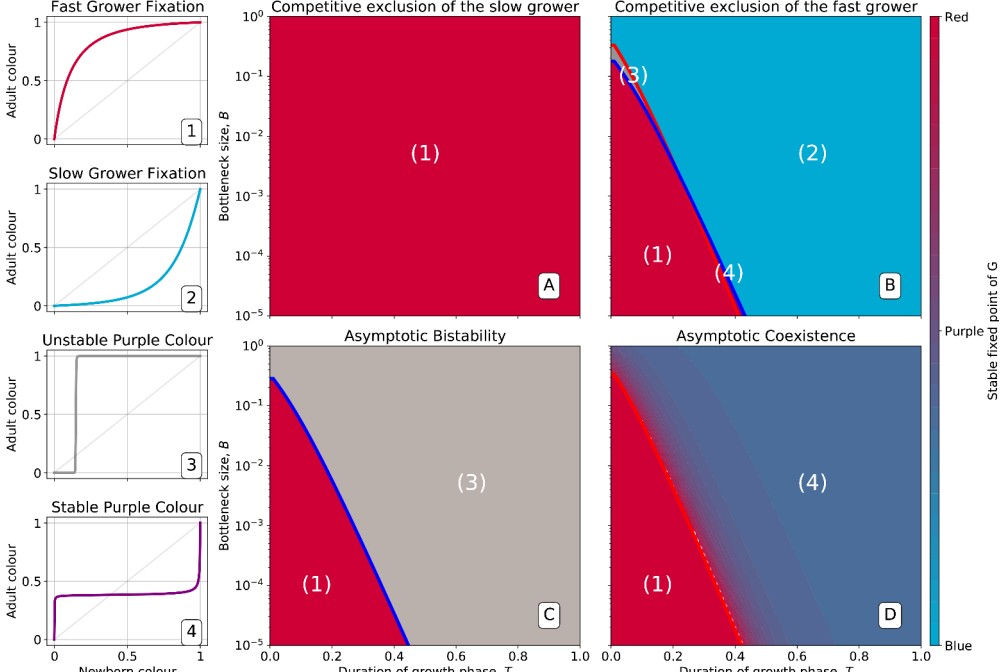

**Appendix 2—figure 3.** Qualitative behaviours of the growth function $G_\theta$. Panels 1–4 represent the possible qualitative shapes, differing in the position and stability of the fixed points, of the growth function $G_\theta$ that approximates the within-collective particle dynamics (orange line in *Figure 5*). (1) $\phi = 1$ is the only stable fixed point (*Figure 5A*), and iteration of the red $G_\theta$ function leads to fixation of red particles. (2) $\phi = 0$ is the only stable fixed point, and iteration of the blue $G_\theta$ function leads to fixation of blue particles. (3) $\phi = 0$ and $\phi = 1$ are two stable fixed points, and iteration of the grey $G_\theta$ function leads to fixation of either red or blue particles depending on initial conditions. (4) $\phi = 0$ and $\phi = 1$ are unstable and there is a stable fixed point between 0 and 1. Iteration of the grey $G_\theta$ function leads to coexistence of both particle types. Panels (**A-D**) show that when red particles are the fast growing types ($r_0 > r_1$), the shape of $G_\theta$ and the position of its fixed points depend on the collective-level parameters $B$ (bottleneck size) and $T$ (growth phase duration). Particle interaction traits generically belong to one of the four intervals (A) $a_{01} < a_{11}$ and $a_{00} < a_{10}$; (B) $a_{11} < a_{01}$ and $a_{10} < a_{00}$; (C) $a_{11} < a_{01}$ and $a_{00} < a_{10}$ ; (D) $a_{01} < a_{11}$ and $a_{10} < a_{00}$ (qualitative nature of the corresponding ecological equilibria is indicated in the titles of the panels, see also Appendix 2). Lines represent the limit of the region of stability of the fixed point of $G_\theta$, as derived by Proposition 4: blue lines for the 'all blue' state $\phi = 0$ and red lines for the 'all red' state $\phi = 1$.

The parameter values that separate regions with qualitatively different fixed points correspond to transcritical bifurcations, where one of the monochromatic fixed points changes its stability. These lines are analytically computed below, thus allowing generalisation of the conclusions drawn from analysing the representative trajectory of *Figure 2*. In the following, we detail analysis of how the fixed points of the $G_\theta$ function depend on particle- and collective-level parameters.

It is worth stressing here that the deterministic model provides a good quantitative approximation of the system with particle-level and collective-level stochasticity provided that fluctuations at both levels are small. This is the case if populations of particles are large (as for instance in the case of bacterial populations) and if mutations of particle traits are rare and of small magnitude. However, in the numerical simulations we performed, the conclusions drawn from ensuing analysis held qualitatively also in the case of large fluctuations.

## Fixed Points of the $G_\theta$ function and their stability

In this paragraph, we list the key properties of the $G_\theta$ function, that determine how the asymptotic collective colour depends on particle and collective parameters. Proofs of the propositions are provided in the following paragraph.

## Proposition 1 (Fixed points of $G_\theta$)

*Let $\theta \in E$ be a set of particle traits, $T \in (0,\infty)$ the duration of the growth phase and $B \in (0,\infty)$ the bottleneck size.*

*Then, $G_\theta(0,T,B) = 0$, and $G_\theta(1,T,B) = 1$, hence $\phi = 0$ and $\phi = 1$ are fixed points of $G_\theta$ $\forall \theta$.*

*Moreover, if the stability with respect to $\phi$ of 0 and 1 is the same, then $G_\theta$ has at least one fixed point $\phi \in (0,1)$.*

The stability of the monochromatic fixed points with respect to the fraction $\phi$ can be numerically assessed. The number and stability of the fixed points as a function of collective-level parameters $B$ and $T$ are illustrated in *Appendix 2—figure 3*. Four different cases are considered, corresponding to the four qualitatively different outcomes of particle-level ecology (competitive exclusion by one or the other type, bistability, coexistence).

The fixed points can be analytically calculated in certain limit cases, corresponding to parameter values when within-collective particle dynamics are exponential or saturating.

## Proposition 2 (Quasi-exponential growth)

*When $T$ is close to 0 and $Ba_{\max} \ll 1$ with $a_{\max} = \max_{i,j} a_{ij}$ the highest element of the competition matrix $\mathbf{A}$, then $G_\theta$ has only two fixed points $\phi = 0$ and $\phi = 1$.*

*Moreover, the monochromatic fixed point $\phi = 1$, corresponding to a population completely composed of the (faster) red type of particle, is stable, whereas the fixed point $\phi = 0$, corresponding to a population composed of the slow growing type of particle, is unstable.*

## Proposition 3 (Saturating growth)

*As $T \to \infty$ the fixed points $\phi^*$ of $G_\theta$ and their stability correspond to the equilibria $x^*$ of **Equation 1**. These equilibria and their stability range are listed below.*

|  | Red alone | Blue alone | Coexistence |
|---|---|---|---|
| Equilibrium $x^*$ | 1 | 0 | $\frac{a_{11}-a_{01}}{Tr(A)-CoTr(A)}$ |
| Stability range | $a_{00}<a_{10}$ | $a_{11}<a_{01}$ | $a_{11}>a_{01}$ and $a_{00}>a_{10}$ |

*Here $Tr(\mathbf{A}) := a_{00} + a_{11}$ is the sum of the diagonal (or trace) of $\mathbf{A}$, and $CoTr(\mathbf{A}) := a_{10} + a_{01}$ denotes the sum of the anti-diagonal elements of $\mathbf{A}$.*

By linearising the system in proximity of the fixed points, it is possible to find exactly the bifurcation parameters where one equilibrium changes stability, thus the limit of the region where there exists an interior fixed point $\phi^*$. The bifurcation values in the space of the collective parameters $T$ and $B$ delimit the region in **Figure 7** where the $G_\theta$ function has an internal fixed point.

Even when the fixed point is different from the optimal value $\hat{\phi}$, it can nonetheless provide a starting point for evolution to optimise collective colour. Extinction of one of the two colours of particles happens instead very rapidly in the region when the monochromatic fixed points are stable, so that collectives have a higher risk of being extinct before inheritance-increasing mutations appear.

## Proposition 4 (Bifurcations of the monochromatic fixed points of $G_\theta$)

*The stability of 0 changes at $(T_B^*, B_B^*)$ and the stability of 1 at $(T_R^*, B_R^*)$ such that:*

$$B_R^* = \frac{e^{-\alpha_1 T_R^*} - e^{r_0 T_R^*}}{a_{00}(1 - e^{-\alpha_1 T_R^*})} \quad \text{with } \alpha_1 = r_0 r_1 \frac{a_{00} - a_{10}}{r_0 a_{00} - r_1 a_{10}}$$

$$B_B^* = \frac{e^{-\alpha_0 T_B^*} - e^{r_1 T_B^*}}{a_{11}(1 - e^{-\alpha_0 T_B^*})} \quad \text{with } \alpha_0 = r_0 r_1 \frac{a_{11} - a_{01}}{r_1 a_{11} - r_0 a_{01}}$$

These results allow understanding of the interplay between time scales of particle-level ecology and collective reproduction, whose relationship changes along an evolutionary trajectory. *Appendix 2—figure 4* illustrates the change of the fixed point of $G_\theta$ with the collective generation duration $T$ for typical particle traits corresponding to the four qualitative classes of asymptotic equilibria for particle ecology. Of particular relevance for understanding the stochastic trajectory illustrated in *Figure 2* is panel D.

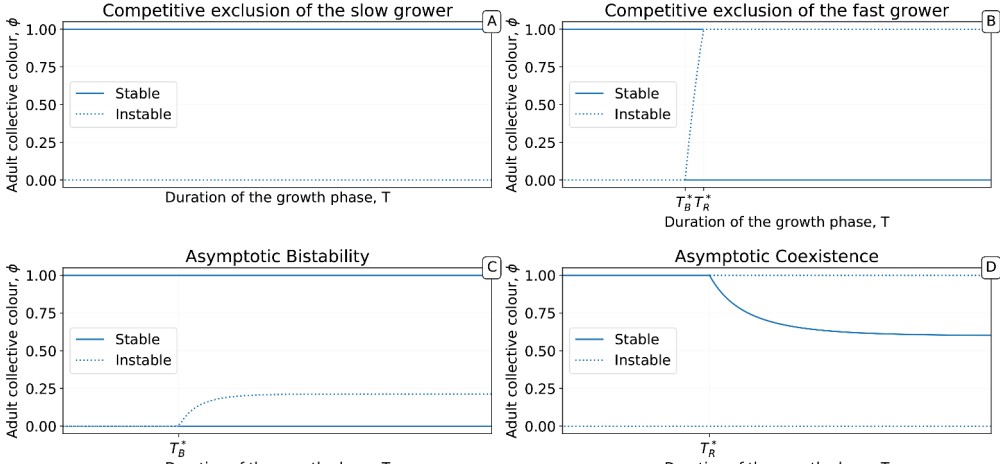

**Appendix 2—figure 4.** Bifurcation diagrams showing the position and stability of the fixed points of $G_\theta$ as a function of the duration of the collective generation $T$. (**A-D**) Particle traits are representative of the scenarios illustrated in *Figure 7A–D*. Of particular interest is the case illustrated in panel $D$, where the $G_\theta$ function acquires — for a sufficiently large separation between the particle maximum division time and the collective generation time — a stable internal fixed point.

When the time scale of exponential particle growth is comparable to $T$, (such as at the beginning of the evolutionary trajectory displayed in *Figure 2*) Proposition 2 indicates that the system is expected to converge to an all-red solution ($\phi = 1$ is the only stable equilibrium). However, at the same time as this fast dynamic occurs, the growth rates change by mutation. Selection during the exponential phase generally favours fast growing mutants, which means that particle populations achieve high-density conditions in a shorter time. The system then effectively behaves as if $T$ had increased, thus leading selection to 'see' interaction traits.

When the time scale of collective reproduction is sufficiently slow with respect to the intra-collective dynamics, the system crosses the bifurcation point $T_R^*$ (Proposition 4), so that the function $G_\theta$ now has an internal fixed point (*Figure 6*). In the stochastic simulations, this means that more collectives are reproducibly found close to the optimal colour. It takes a relatively short time to adjust the particle traits so that the fixed point is close to the optimum $\widehat{\phi}$. In this case, the deterministic approximation produces a close to perfect inheritance of the collective colour. However, fluctuations in particle numbers and in composition at birth still result in a large variance of colours among collectives in the stochastic system.

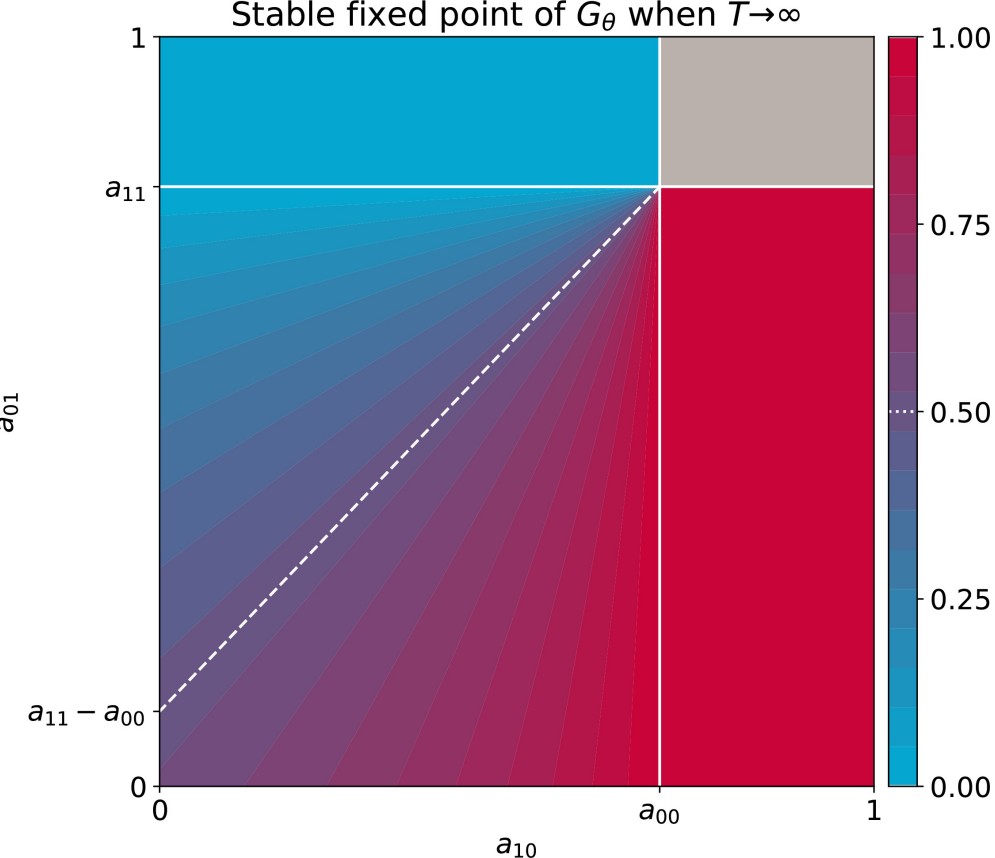

**Appendix 2—figure 5.** Stable equilibria of the particle ecology as a function of inter-colour competition traits. These equilibria correspond to the limit of the fixed points of the $G_\theta$ function when particle-level and collective-level time scales are well separated ( $r \gg \frac{1}{T}$), derived in proposition 3. Other interaction parameters are $a_{00} = 0.7$ and $a_{11} = 0.8$, and the result is independent of growth rates. The grey area indicates bistability.

Here, starts the last and slowest phase of the evolutionary trajectory, which results in colour variance reduction through improvement of the ability of particles to correct variations in colour. This is achieved by attaining faster the particle ecological equilibrium, so that fluctuations are more efficiently dampened by demographic dynamics. As a consequence, the conditions described by Proposition 3 will be met. This allows identification of a surface in parameter space, where the fixed point of the $G_\theta$ function $\phi^*$ identifies with the ecological equilibrium $x^*$, that contains evolutionary equilibria. The ecological equilibrium is displayed in *Appendix 2—figure 4* as a function of the cross-colour interaction parameters $a_{01}$ and $a_{10}$. In the regime where particle and collective time scales are well separated ($r \gg 1/T$), then interaction parameters that correspond to the optimal colour satisfy the following relationship:

$$a_{00} - a_{10} + \left(1 - \frac{1}{x^*}\right)(a_{11} - a_{01}) = 0. \tag{4}$$

This relation identifies the white dotted line in *Appendix 2—figure 4* (and in *Figure 8B*). Once it is attained, mutations cause the deterministic system to move neutrally on this surface. As the stochastic simulation shows, particle parameters keep evolving directionally so as to reduce phenotypic variance. This is achieved by making $G_\theta$ increasingly flatter in the vicinity of the fixed point, so that the target colour is not only more stable, but it is reached in fewer collective generations.

Successive events of mutation and substitution progressively lead to a growing asymmetry in the ecological relationship between the two types of particles: that with smaller carrying capacity becomes insensitive to the other colour; the latter instead experiences competition, so that its growth is curbed and optimal proportion of colours is eventually realized (see *Figure 3D*). As

illustrated by *Appendix 2—figure 6*, this conclusion is independent of what is the maximum growth rate, that only affects the advantage of one type at initial stages of growth. Indeed, the position on the intercept between the optimality line and the y axis in *Appendix 2—figure 5* and *Appendix 2—figure 6* only depends on the difference between intra-type competition parameters. A consequence of this is that fast-growing types systematically display, when they are also those with a larger carrying capacity, a population overshoot.

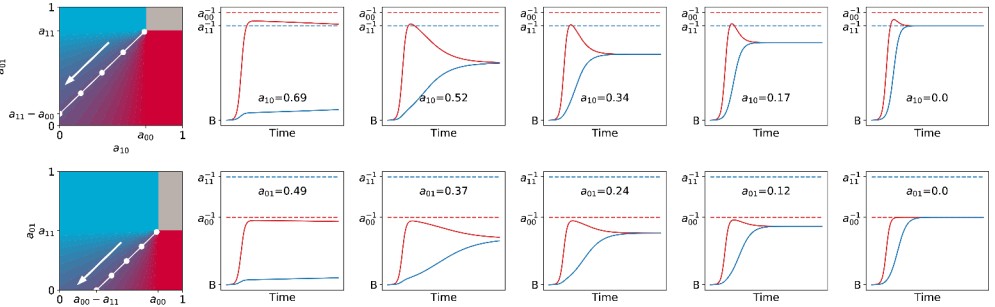

**Appendix 2—figure 6.** Asymmetric interaction ensures fastest convergence toward the ecological equilibrium. Particle population dynamics are illustrated for increasingly asymmetric values of $(a_{01}, a_{10})$, keeping the ecological equilibrium fix at $x^* = 0.5$ (along white manifold, in the direction of the arrow). The top panels correspond to cases when the faster-growing particles have a higher carrying capacity $((a_{00}, a_{11}) = (0.7, 0.8))$, the bottom panels to the opposite $((a_{00}, a_{11}) = (0.8, 0.5))$. In both cases $r_0 = 60$, $r_1 = 40$, $T = 1$, $B = 0.001$.

## Proofs

In this section, we present the proof of propositions 2–4 above.

### Proof of Proposition 1

In the $(N, x)$ coordinates we have seen that $\frac{df(t)}{dt} = g(x(t), N(t))$ and that $x(t) = 0$ and $x(t) = 1$ are trivial roots of the polynomial $g$ (*Equation 2*). Hence 0 and 1 are always fixed points of $G_\theta$.

Since $g$ and $h$ (from *Equation 2*) are polynomials of $(N, x)$, they are smooth (of differentiability class $C_\infty$) on $\mathbb{R}^2$. Thus, the global flow corresponding to the Cauchy problem is also smooth on $\mathbb{R}^2$ is the partial application of the global flow to the case where $N_0 = B$. Therefore, $G_\theta$ is continuous on $[0, 1]$.

Moreover, let us suppose that 0 and 1 are both unstable. Then $G'_\theta(0, T, B) > 1$ and $G'_\theta(1, T, B) > 1$, with $G'_\theta$ the derivative of $G_\theta$ with respect to its first variable. As a consequence there is an $\varepsilon \in \mathbb{R}$ such that $G_\theta(\varepsilon, T, B) > \varepsilon$ and $G_\theta(1 - \varepsilon, T, B) < 1 - \varepsilon$. Since $G_\theta$ is continuous, there is at least one $c \in [0, 1]$ such that $G_\theta(c, T, B) = c$ by virtue of the intermediate value theorem.

In practice, we never encountered cases when more than one internal fixed point was present. However multistability is expected to occur if the equations describing particle ecology had higher-order nonlinearities.

### Proof of Proposition 2

Since the nonlinear terms in *Equation 3* are smaller than $Ba_{\max}$, and this is negligible with respect to 1, particle ecology is approximated by its linearization as long as the population size remains close to the bottleneck value. Around $t = 0$, the Cauchy problem can be written as:

$$\begin{cases} \frac{dN_0}{dt} = r_0 B x \\ \frac{dN_1}{dt} = r_1 B (1 - x) \\ N_0(0) = x B \\ N_1(0) = (1 - x) B \end{cases}$$

In the $(N, x)$ coordinates, the total population size grows exponentially and is decoupled from the colour. On the other hand, $x(t)$ follows the logistic differential equation:

$$\frac{dx}{dt} = \frac{d}{dt}\frac{N_0}{N_0 + N_1} = (r_0 - r_1)x(1 - x) \tag{5}$$

which can be integrated.

For $T$ sufficiently small for the population to be in exponential growth phase, the growth map $G_\theta$ can be approximated by the solution $\widetilde{G}_\theta$ of *Equation 5*:

$$G_\theta(x, T, B) \approx \frac{1}{1 + \left(\frac{1}{x} - 1\right)e^{-(r_0 - r_1)T}} := \widetilde{G}_\theta(x, T, B)$$

This function is strictly convex (or concave) on (0, 1) depending on the sign of $r_0 - r_1$:

$$\frac{\partial^2 \widetilde{G}_\theta(x, T, B)}{\partial x^2} = \frac{2\left[1 - e^{(r_0 - r_1)t}\right]}{\left(xe^{(r_0 - r_1)t} + 1 - x\right)^3}$$

Since $0 < x < 1$ and $t > 0$, $\frac{\partial^2 \widetilde{G}_\theta(x,T)}{\partial x^2}$ is of the same sign as $1 - e^{t(r_0 - r_1)}$, that is strictly positive if $r_1 > r_0$, or strictly negative if $r_0 > r_1$.

Thus, $\widetilde{G}$ is strictly convex on $(0, 1)$ if $r_1 > r_0$, and strictly concave on $(0, 1)$ if $r_0 > r_1$. In the first case, red colour $x = 0$ is an unstable equilibrium and blue colour $x = 1$ a stable one, and vice-versa in the second case. Note that the segment $s = [(0, 0), (1, 1)]$ is a chord of $\widetilde{G}$. Therefore, the strictly convex (resp. concave) $\widetilde{G}$ do not intersect $s$ except in $(0, 0)$ and $(1, 1)$.

## Proof of Proposition 3

When the collective generation time $T$ is much longer than the demographic time scale, the populations within droplets at the adult stage are well approximated by the equilibrium solution of the Lotka-Volterra *Equation 1*. Solving simultaneously equations $\frac{dN_0}{dt} = 0$ and $\frac{dN_1}{dt} = 0$ (or equivalently $\frac{dN}{dt} = 0, \frac{dx}{dt} = 0$) yields the four equilibria listed below in both coordinate systems. Linear stability analysis allows one to determine the parameter intervals where these are stable, listed below.

| Name | Red alone | Blue alone | Coexistence | Extinction |
|---|---|---|---|---|
| Position in $[N_0, N_1]$ | $[\frac{1}{a_{00}}, 0]$ | $[0, \frac{1}{a_{11}}]$ | $[\frac{a_{11} - a_{01}}{det(A)}, \frac{a_{00} - a_{10}}{det(A)}]$ | $[0, 0]$ |
| Position in $[N, f]$ | $[\frac{1}{a_{00}}, 1]$ | $[\frac{1}{a_{11}}, 0]$ | $[\frac{Tr(A) - CoTr(A)}{det(A)}, \frac{a_{11} - a_{01}}{Tr(A) - CoTr(A)}]$ | Undefined |
| Condition for stability | $a_{00} < a_{10}$ | $a_{11} < a_{01}$ | $a_{11} > a_{01}$ and $a_{00} > a_{10}$ | Never |

## Proof of Proposition 4

We consider the case when the fixed point 0 changes stability. The stability of the fixed point 1 can be studied analogously.

We aim at identifying the values of the collective parameters $(T, B)$ where a fixed point with $x = 0$ changes stability through a transcritical bifurcation. The difficulty lies in the fact that one needs to estimate the dynamics of the second variable $N$ in order to study the stability of the 2-D system. Luckily, this can be done in the limit when the collective contains almost exclusively particles of one single colour (in this case, blue).

Total population size is in this case decoupled from colour and *Equation 2* can be integrated with initial condition $(N_0 x_0) = (B, 0)$, yielding the following trajectory:

$$\widetilde{N}(t, B) = \frac{1}{a_{11} - \left(a_{11} - \frac{1}{B}\right)e^{-r_1 t}}$$

As long as $x$ is small, the time derivative is approximated by the non-autonomous system:

$$\frac{dx}{dt} \approx xh(x, \widetilde{N}(t, B))$$

with $h$ as defined in *Equation 2*:

$$h(0,\widetilde{N}(t,B)) = (r_0 - r_1) + \widetilde{N}(t,B)(a_{11}r_1 - a_{01}r_0)$$

Solving this equation allows computation of the adult colour as a function of the parameters $T$ and $B$. At the bifurcation point $(T^*, B^*)$, where stability of the 0 fixed point changes, the newborn colour is the same as the adult colour: $x(T^*, B^*) = x(0, B^*)$. $(T^*, B^*)$ are then solutions of the integral equation:

$$0 = \int_0^{T^*} h(0, \widetilde{N}(s, B^*)) ds$$

$$0 = (r_0 - r_1) + (a_{11}r_1 - a_{01}r_0) \int_0^{T^*} \widetilde{N}(s, B^*) ds$$

$$0 = T^* r_0 \left(1 - \frac{a_{01}}{a_{11}}\right) + \left(1 - \frac{r_0 a_{01}}{r_1 a_{11}}\right) \ln\left(\frac{B^* a_{11} + e^{-r_1 T^*}}{B^* a_{11} + 1}\right)$$

Solving for $B^*$ we get:

$$B^* = \frac{e^{-\alpha T^*} - e^{r_0 T^*}}{a_{00}(1 - e^{-\alpha T^*})}$$

With $\alpha = r_0 r_1 \frac{a_{00} - a_{10}}{r_0 a_{00} - r_1 a_{10}}$

*Figure 7* shows that this approximation retrieves accurately the numerically computed bifurcation lines.

## Appendix 3

### Beyond competitive interactions

In the main text, the model is limited to deleterious (competitive) interactions, whereby the inter-colour interactions are such that one colour always has a negative density-dependent effect on the growth of the other colour. This choice reflects the expectation that, as in the example of the droplet experiment, competition would be the main driver of interactions among two previously unconnected bacterial species growing on a common resource.

However, it is interesting to consider what happens if interactions of other types are taken into account. This section presents two additional interaction classes: exploitative (e.g. predator-prey or parasitic) interactions ($a_{01}<0<a_{10}$ or $a_{10}<0<a_{01}$), and mutualism ($a_{01}<0$ and $a_{10}<0$). Note that exploitative interactions correspond to non-obligate predation (or parasitism): the predator (or the parasite) can sustain a non-null population density even in the absence of the prey (or host). Obligate interactions can be modelled in the limit where the carrying capacity of the exploiters tends to zero. Similarly, unless the carrying capacities of both types tend to zero, mutualism is also not obligate.

The deterministic model (Appendix 2 *Equation 1*) can be straightforwardly extended to these larger parameter ranges, allowing prediction of the outcome of the evolutionary dynamics. *Appendix 2—figure 5* presents the position of the stable fixed point of $G_\theta$ for negative inter-colour interaction parameters when the time scale of collective reproduction is sufficiently slow with respect to the intra-collective dynamic $\left(r \gg \frac{1}{T}\right)$. *Appendix 3—figure 1* represents the same fixed point for extended values of the interaction parameters. Each quadrant corresponds to a different kind of ecological interaction (competition, exploitation and mutualism). While competition is discussed in the main text, other kinds of interactions are discussed below.

Whenever evolution leads to separation in time-scales between particle and collective dynamics (typically by increasing particle growth rates), the evolutionary dynamic within each quadrant will be akin to the competition case: first, successive invasions of mutants bring the system on the manifold of interaction parameters where the equilibrium colour $x^*$ corresponds to the selected colour $\widehat{\phi}$ (Appendix 2 *Equation 4*); second, the system evolves on this manifold toward regions of faster ecological convergence.

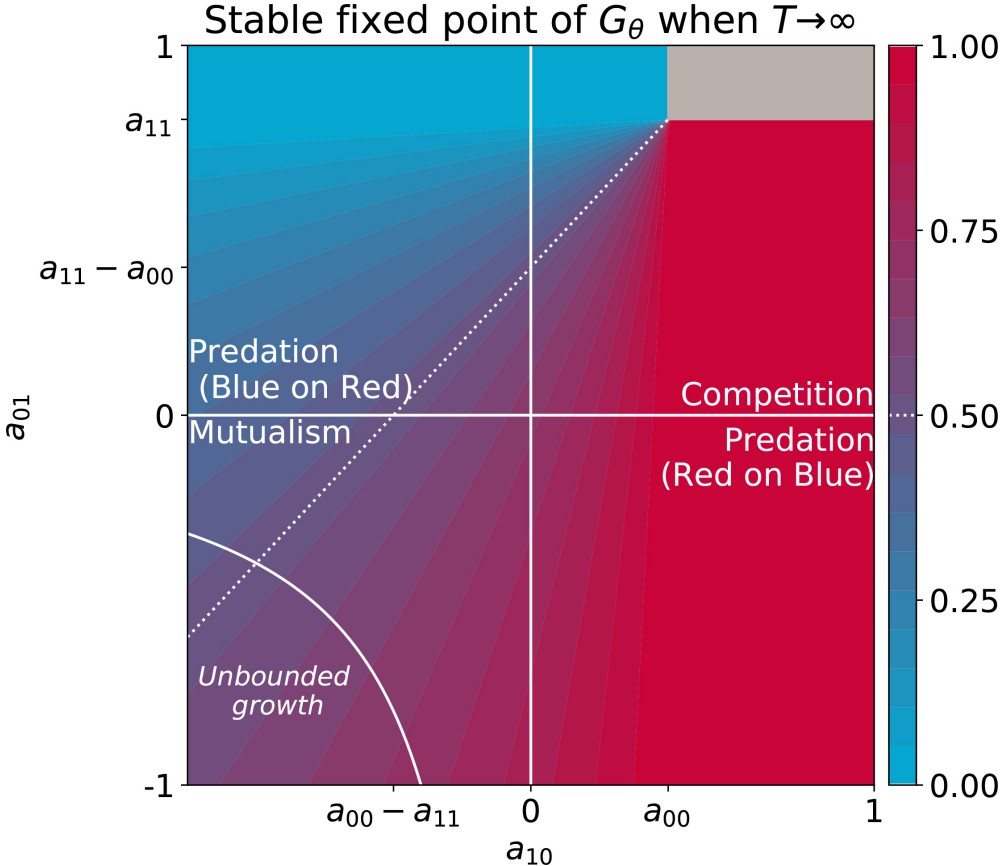

**Appendix 3—figure 1.** Asymptotic colour of the particle ecology as a function of inter-colour competition traits. These equilibria correspond to the limit of the fixed points of the $G_\theta$ function when particle-level and collective-level time scales are well separated ( $r \gg \frac{1}{T}$ ), derived in proposition 3. Other interaction parameters are $a_{00} = 0.7$ and $a_{11} = 0.8$, and the result is independent of growth rates. The grey area indicates bistability. This figure extends *Appendix 2—figure 5*.

### Exploitative interactions

Consider first that $a_{01} < 0 < a_{10}$, so that red individuals (type 0) are the exploiters (predators or parasites) and blue individuals (type 1) are the exploited (prey or hosts). This corresponds to the bottom-right quadrant of *Appendix 3—figure 1*.

When the exploiter competes with the exploited more than with other exploiters ($a_{00} < a_{10}$), the stable equilibrium for Appendix 2 *Equation 1* is monochromatic red (see Appendix 2 Proposition 3). However, when the the competitive interaction is stronger among exploiters ($a_{00} > a_{10}$), the coexistence equilibrium is stable and correspond to a colour of $\frac{a_{11} - a_{01}}{Tr(A) - CoTr(A)}$ (see Appendix 2 Proposition 3). In the coexistence region of the bottom-left quadrant, the collective colour can take up values in the range $\left[\frac{a_{11}}{a_{11} + a_{00}}, 1\right]$ bounded on one side by the extinction of the exploited type, and on the other side by the ratio of carrying capacity obtained when there is no competitive interactions. This range does not contain the target colour $\widehat{\phi} = 0.5$ in the example illustrated in *Appendix 3—figure 1* because the exploiter has a higher carrying capacity than the exploited $\left(\frac{1}{a_{11}} < \frac{1}{a_{00}}\right)$. It is thus impossible to achieve the target colour $\widehat{\phi} = 0.5$ in the lower right quadrant. Unless the nature of ecological interactions changes, over evolutionary times the population of collectives will move towards the absence of interactions. Here, collectives of colour close to the optimum can still manifest in the stochastic individual-based model as the effect of sampling at dilution and fluctuations during growth, but the average colour will remain off target.

In the upper left quadrant, where $a_{10} < 0 < a_{01}$, blue individuals (type 0) are the exploiters (predators or parasites) and red individuals are the exploited (prey or hosts). In the coexistence region (i.e.,

when the competition is stronger among exploiters than between exploiters and exploited $a_{01} < a_{11}$), the collective colour can take up values in the range $\left[0, \frac{a_{11}}{a_{11}+a_{00}}\right]$, that contains the target $\widehat{\phi} = 0.5$ in the example illustrated in *Appendix 3—figure 1* because the exploiter has a lower carrying capacity than the exploited $\left(\frac{1}{a_{11}} < \frac{1}{a_{00}}\right)$. Within the boundaries of this quadrant, long-term evolution will tend to reduce the (positive) interaction $a_{01}$, thus mitigating exploitation, while at the same time making the (negative) interaction term $a_{10}$ smaller, that is increasing the advantage provided by the exploited to the exploiter. Again, if the nature of ecological interactions does not change, the system will evolve towards a case where one particle type (in this case, the exploiter, that also has smaller carrying capacity), gets decoupled from the demography of the other type.

## Mutualistic interactions

Consider now that red and blue individuals have non-obligate, mutualistic interactions. This corresponds to both inter-colour competition traits $a_{01}$ and $a_{10}$ being negative, and can be visualised in the bottom-left quadrant of *Appendix 3—figure 1*.

When interactions are mutualistic, the equilibria of Appendix 2 *Equation 1* resulting from the extinction of one type or the other are never stable ($a_{00} < a_{10}$ or $a_{11} < a_{01}$ are never fulfilled since $a_{01} < 0 < a_{00}$ and $a_{01} < 0 < a_{11}$ respectively, see Appendix 2 Proposition 3) and the solution is always coexistence.

Depending on the sign of the determinant of the interaction matrix $\mathbf{A}$, the dynamics within a collective generation can have two qualitatively different behaviours. When mutualistic interactions have small intensity, the intra-collective dynamics attains an equilibrium, even though the population size of one of the two particle types is larger than their respective carrying capacities (because of the positive contribution of the other type). However, when $\frac{a_{00}a_{11}}{a_{01}a_{10}} < 1$ (white line in *Appendix 3—figure 1*), the populations would undergo unbounded growth if they were not embedded in collectives of finite lifetime.

This unrealistic feature of simplified Lotka-Volterra models, that makes them not well fit to describe mutualistic interactions, is not particularly problematic in our case, since the fraction $x$ of red particles converges to its equilibrium value even though the population size $N$ diverges. *Appendix 3—figure 1* illustrates how the within-collective particle dynamics changes along an evolutionary trajectory towards increasingly strong mutualistic interactions. If no physical or biological constraint keeps parameters in the region of bounded growth, then adherence of the model predictions from reality would need to be tested by studying their robustness to inclusion of other sources of co-limitation, such as, for instance, nutrient exhaustion.

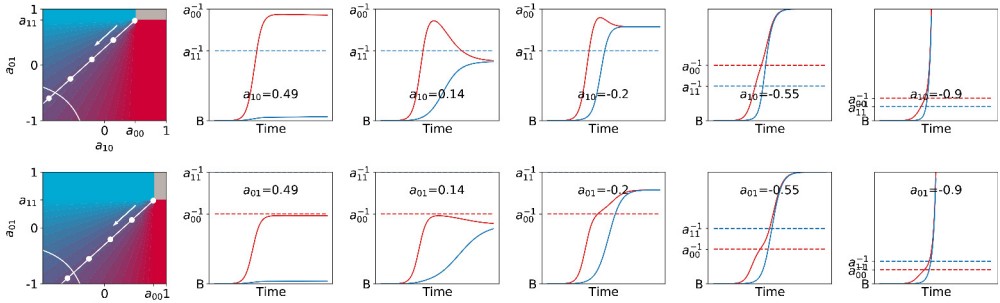

**Appendix 3—figure 2.** Mutualistic interaction can lead to non-saturating population growth. Particle population dynamics are illustrated for increasingly mutualistic values of $(a_{01}, a_{10})$, keeping the ecological equilibrium fixed at $x^* = 0.5$ (moving along the white manifold, in the direction of the arrow). When the determinant of $\mathbf{A}$ is negative, population growth is unbounded in the limit of infinitely long duration of the collective generation. As long as $T$ is finite, however, the adult proportion of red and blue particles can be reached even though both populations grow exponentially (with growth rates that depend on the interaction parameters). The top panels correspond to cases when the faster growing particles have a higher carrying capacity ($(a_{00}, a_{11}) = (0.5, 0.8)$), the bottom panels is the opposite ($(a_{00}, a_{11}) = (0.8, 0.5)$). In both cases $r_0 = 30$, $r_1 = 30$, $T = 1$, $B = 0.001$.

## Transition between classes of ecological interactions

So far, we have examined different ecological interactions separately, as they are usually considered to belong to separate categories. It is, however, conceivable that mutations induce a competitive interaction to become exploitative, or an exploitative interaction to become mutualistic (*Sørensen et al., 2019*), which in our case would be the consequence of the inter-colour interaction parameters changing sign. In this case, the prediction of the deterministic model is that, as long as there are no constraints that limit the variation of the interaction parameters, selection at the collective level should lead particles to become more and more benign towards each other, and eventually reach a regime where their exchanges are mutualistic.

