## [Decision Letter]

**Acceptance summary:**

Doulcier et al. tackled multi-level community selection: selection at the level of individual cells for fast growth and at the level of whole communities for a desired community property. The authors showed that when selecting communities based on species ratios, the growth rates and the inter-species competition effects can evolve such that a desired species ratio (here 1:1) can be achieved. This work contributes to a growing interest in using multi-level selection to achieve desired community properties.

**Decision letter after peer review:**

Thank you for submitting your article "Eco-evolutionary dynamics of nested Darwinian populations and the emergence of community-level heredity" for consideration by *eLife*. Your article has been reviewed by three peer reviewers, including Wenying Shou as the Reviewing Editor and Reviewer #1, and the evaluation has been overseen by Patricia Wittkopp as the Senior Editor. The following individuals involved in review of your submission have agreed to reveal their identity:); Sara Mitri (Reviewer #2); Alvaro Sanchez (Reviewer #3).

The reviewers have discussed the reviews with one another and the Reviewing Editor has drafted this decision to help you prepare a revised submission.

Summary:

Doulcier et al. tackled multi-level community selection: selection at the level of individual cells for fast growth and at the level of whole communities for a desired community property. The authors showed that when selecting communities based on species ratios, the growth rates and the inter-species competition effects can evolve such that a desired species ratio (here 1:1) can be achieved. The evolved mechanism is rather intuitive: species Blue grows faster than species Red, but species Blue is subject to competitive inhibition by species Red. This allows the species ratio to be 1:1. Consistent with intuition, when possibilities for competition coefficients are eliminated, growth rates of the two species evolved to be equal. When basal grow rates were not allowed to evolve, asymmetric competition arose to achieve equal post-competition growth rates. All three reviewers are positive about this work.

Essential revisions:

1) Allowing ecological interactions to evolve to be positive.

2) Statistics of simulation repeats.

3) What happens if selecting for uneven ratios when noise is present?

4) What does "rapidly" really mean?

To help you revise, the original reviews are attached.

Reviewer 1:

1) Define "interactions".

2) "Developmental correction" is a biologist's way of saying "steady-state species ratio" or "attractor/fixed point". There are multiple ways of achieving steady-state species ratios. The authors had limited species interactions to competition (non-positive coefficients). If they allow interactions to change signs, then commensal or mutually beneficial interactions can arise to stabilize strain ratios. "Developmental correction thus selects for maximal asymmetry in interactions" may no longer hold for positive interactions. This needs to be checked.

Reviewer #2:

In this manuscript, the authors set out to explore whether it's possible to select for phenotypes at the community level, that arise in spite of competition between community members at the individual level. Using a computational model, they show that this is indeed possible, but relies on communities being properly enclosed and only passing on their individuals to their own offspring communities. They also analyse the strategy allowing for the evolution of community phenotypes. They show that community members evolve individual phenotypes that allow them to robustly converge to the community property under selection, independently of the initial community state. They posit that a similar mechanism might explain major evolutionary transitions, allowing lower level entities to be selected as whole entities.

Overall, we like the manuscript. It makes, in our opinion, a major contribution in understanding how competition between individual community members can be overcome – or even used – to allow for selection of the community phenotype. The idea that selection can favor individual-level parameters that have a functional form that increases resemblance between parent and offspring and robustness to initial conditions is very interesting and overcomes some of the issues with community selection observed in previous studies (Xie et al). The authors also nicely show that once this strategy has evolved, it is ecologically stable, and they explore how the evolutionary trajectory depends on starting conditions.

We have some general thoughts that we think are worth discussing further in the manuscript.

First, what we find was missing were statistics on what happens on different runs of the simulation. As far as we could tell, all data shown is based on individual "example" runs. The authors claim that these are general, but there is no data to support that. Please add statistics on the repeatability of the results. A related point: is the outcome always qualitatively similar? Specifically that a) the faster grower becomes less competitive with respect to growth rate, and b) that the emerging population regulation consists of the slower grower suppressing the faster grower while c) the faster grower does not affect the slower. It was not clear to us why, and we believe this is important to understanding the conclusion of the paper.

Second, we believe the authors could elaborate more on the specificity of their system and how their findings depend on them. It is not clear to what extent the same result would be observed in a different system that (1) has more than 2 types, (2) where selection acts on total abundance rather than ratios of two colors, (3) where the selective regime is different (the authors say "A non-exhaustive exploration of other selection rules indicates that the qualitative results of the model are robust to changes in the selective regime, as long as collectives with an optimal colour are favoured and the collective population does not go extinct." It would be worth expanding on this in the supplement); and finally, (4) where the phenotype being selected for is not intermediate. Would this work equally well if the phenotype selected is almost blue or almost red? In short, under what general conditions do you expect to see the same?

From the ecological literature, it is known that coexistence between two competing types requires some limiting force for the strongest grower (e.g. predator). Is that simply what you are selecting for? It may be good to make a link to this literature.

We are also unsure about the parallels made to developmental processes and think this would merit some clarification. Does one see this strategy where a slower-growing type limits the growth of a faster-growing one? Or are the authors hypothesizing that this might explain how smooth developmental processes occur?

Finally, Figure 2 is lacking in visual clarity (color scheme). The selection method should also be better explained in the main text.

Please note that we did not work through all the math nor run the code.

Reviewer #3:

I very much enjoyed this paper. I do not have any significant concerns.

[Editors' note: further revisions were suggested prior to acceptance, as described below.]

Thank you for re-submitting your article "Eco-evolutionary dynamics of nested Darwinian populations and the emergence of community-level heredity" for consideration by *eLife*. Your article has been re-reviewed by two of the three original reviewers, and the evaluation has been overseen by a Reviewing Editor and Patricia Wittkopp as the Senior Editor.

The only thing that we believe is still missing is a paragraph in the Discussion on the generality of these findings. We understand that adding more than 2 types, trying different selection regimes or selecting on total abundance is beyond the scope of the current manuscript, but we still believe these questions may remain for a reader of the paper, so it is worth bringing them up in the Discussion as future directions.

---

## [Author Response]

Essential revisions:1) Allowing ecological interactions to evolve to be positive.

We have performed further analyses in which interactions evolve to be positive. This is included in a new supplementary document (Appendix 3) and mentioned in the text.

Since the deterministic approximation for the intra-collective particle population dynamics is not dependent on the sign of the interaction rates, we were able to use the steady-state of the G function to study how colour can reach its target value by evolution of particle traits. In Appendix 3 we report the existence (or not) of evolutionary solutions for three additional classes of ecological interaction: exploitation of the type that has larger or smaller carrying capacity (interaction parameters of opposite sign), and mutualism (positive interaction parameters).

We firstly considered the four classes of interaction separately, but show that if interaction parameters are free to change sign, maximal asymmetry of competitive interactions will be replaced by exploitative interactions first, then by mutualism of increasing strength, as correctly intuited by reviewer 1.

We believe that in these cases, even more than for competitive interactions, constrains on particle-level parameters should play a fundamental role in determining the solutions that are actually achievable. For instance, increasing mutualism would lead to the urrealistic situation – typical of Lotka-Volterra models – where mutual benefits are so high that the two evolved types would, in the absence of the imposed collective life cycle, grow indefinitely.

We have commented on this point in the Discussion, where we refer to the new Appendix 3, and introduced earlier the concept that different types of ecological interactions can be conceived (Results subsection “A nested model of collective evolution”).

2) Statistics of simulation repeats.

We perfectly agree with the reviewers that the extent to which our reference simulation is representative of other realizations of the same stochastic process needed to be quantified. As mentioned above, the computational demands associated with such simulations limits the possibility of obtaining this information systematically for any parameter value. We thus realized multiple (20) simulations by fixing the non-evolving carrying capacities to values that illustrate different regions in the space of ecological interactions.

We have expanded Appendix 1 so as to include statistics of the variation of the target colour at the end of the simulations, together with the corresponding variation of the underlying inter-colour interaction and maximal growth parameters for the two particle types of different colour. As already observed when discussing the persistence of the evolutionary trajectory, at the evolutionary equilibrium growth rate becomes largely irrelevant (as long as it is large enough), and therefore is more variable than inter-colour interaction parameters.

We have expanded the first section of the Results in order to include this information and refer to the new paragraph “Variability of the derived particle traits” within Appendix 1.

3) What happens if selecting for uneven ratios when noise is present?

By looking at the deterministic model (Appendix 2—figure 5), it is clear that in the absence of individual-level stochasticity the target colour influences the relative values of the evolving traits and also the resilience of the fixed point to fluctuations (given by the gradient of the fixed point position). However, we expect that fluctuations associated notably with sampling might have major effects on the evolutionary trajectory.

In order to test the effect of the stochastic individual-based process, we ran multiple (10) simulations for different levels of the target colour and expanded Appendix 1 accordingly (paragraph “Different target colours”). As shown in the new Appendix 1—figure 3, the target colour is always successfully achieved, with variation among independent simulations being slightly larger for more extreme colours. However, the variation in the corresponding particle traits is broader, and increases for more extreme target values, together with the time scale associated to reaching the evolutionary equilibrium. These additional observations have been mentioned in the main text, subsection “Selection on collectives drives the evolution of particle traits”, where we refer to the new paragraph “Different target colours” within Appendix 1.

4) What does "rapidly" really mean?

We realise that this term may appear uninformative when addressing the neutral collective selection regime. This was mostly a consequence of mention of issues pertaining to timescales, prior to the later more complete elaboration. Consequently we have removed this early mention and added later that the time scale of early colour variation under collective selection, driven by differences in growth rate, is comparable to that observed in the neutral regime.

We have also added a term of comparison with the stochastic corrector mechanism, that acts on constant particle traits, therefore in conditions where ecological dynamics are dominated by differences in maximal growth rate (Appendix 1—figure 5). The evolutionary dynamics in this case explains the transient regime on short time scales, when the average colour is still very different from the target.

Taking also into account some of the detailed comments, we have expanded the description of the numerical simulation in subsection “Selection on collectives drives the evolution of particle traits”.

To help you revise, the original reviews are attached.Reviewer 1:1) Define "interactions".

We now define interactions in the second paragraph of the Results, where we also mention the possibility of having different kinds of ecological interactions other than competition.

2) "Developmental correction" is a biologist's way of saying "steady-state species ratio" or "attractor/fixed point". There are multiple ways of achieving steady-state species ratios.

The steady state of the growth function corresponds to a trajectory connecting the state of a collective at birth to that at adult stage, and its value does not necessarily correspond to the equilibrium of the particle population dynamics (rather, it captures some property of the transient). We therefore consider that it has many analogies to developmental processes. Moreover, we contrast it with stochastic correction, where correction is not determined by particle ecology that primes, but by fluctuations due to the sampling. We feel therefore that the metaphor is useful to convey our message.

The authors had limited species interactions to competition (non-positive coefficients). If they allow interactions to change signs, then commensal or mutually beneficial interactions can arise to stabilize strain ratios. "Developmental correction thus selects for maximal asymmetry in interactions" may no longer hold for positive interactions. This needs to be checked.

See Essential revision #1.

Reviewer #2:[…]First, what we find was missing were statistics on what happens on different runs of the simulation. As far as we could tell, all data shown is based on individual "example" runs. The authors claim that these are general, but there is no data to support that. Please add statistics on the repeatability of the results. A related point: is the outcome always qualitatively similar? Specifically that a) the faster grower becomes less competitive with respect to growth rate, and b) that the emerging population regulation consists of the slower grower suppressing the faster grower while c) the faster grower does not affect the slower. It was not clear to us why, and we believe this is important to understanding the conclusion of the paper.

See Essential revision #2.

Second, we believe the authors could elaborate more on the specificity of their system and how their findings depend on them. It is not clear to what extent the same result would be observed in a different system that (1) has more than 2 types, (2) where selection acts on total abundance rather than ratios of two colors, (3) where the selective regime is different (the authors say "A non-exhaustive exploration of other selection rules indicates that the qualitative results of the model are robust to changes in the selective regime, as long as collectives with an optimal colour are favoured and the collective population does not go extinct." It would be worth expanding on this in the supplement); and finally, (4) where the phenotype being selected for is not intermediate. Would this work equally well if the phenotype selected is almost blue or almost red? In short, under what general conditions do you expect to see the same?

There are many interesting possible extensions of this model, and we have explored just a few, maintaining the basic assumptions that only two colours are present in the population and that optimal collective function requires both colours to be represented (whereas yield maximization can also be obtained by one type alone).

After having tried different selection schemes and realized that they yielded similar results (as long as the collective population did not go extinct), we chose a selection rule that was simple to implement, both in the individual-based numerical model and in possible applications. A systematic exploration of how different selection rules affect the probability of reaching the target colour within a finite time would however demand a much larger study that goes beyond the scope of this work.

However, as noted in response to Essential revisions #1 and #3, we have investigated the effects of changes in interaction rate and target colour.

From the ecological literature, it is known that coexistence between two competing types requires some limiting force for the strongest grower (e.g. predator). Is that simply what you are selecting for? It may be good to make a link to this literature.

The competitive Lotka-Volterra equations are indeed widely used in ecology. They are so widespread that we decided that no reference to specific literature was warranted.

If the “strongest grower” is defined based on its exponential growth rate r, then it is important to note that in the derived population not this parameter, but the interaction traits, are the chief determinants of the adult colour. See our response to Essential revision #2.

We are also unsure about the parallels made to developmental processes and think this would merit some clarification. Does one see this strategy where a slower-growing type limits the growth of a faster-growing one? Or are the authors hypothesizing that this might explain how smooth developmental processes occur?

Selection of the optimal colour, acting on adult collectives, changes the continuous dynamics that link, within collectives, the newborn to the adult state. In particular, this dynamic is increasingly “canalized”, as illustrated in Figure 3. We have improved Figure 3 in order to make this concept clearer.

Finally, Figure 2 is lacking in visual clarity (color scheme). The selection method should also be better explained in the main text.

We have improved presentation of the model in the main text (following the detailed comments) and the colour contrast of Figure 2. However, we refer the reader to Appendix 1 for all the details on implementation of the selective process. On the basis of the deterministic system and on non-exhaustive exploration of alternative selection schemes, we believe that those details are not essential to understand the key messages of this study.

[Editors' note: further revisions were suggested prior to acceptance, as described below.]

The only thing that we believe is still missing is a paragraph in the Discussion on the generality of these findings. We understand that adding more than 2 types, trying different selection regimes or selecting on total abundance is beyond the scope of the current manuscript, but we still believe these questions may remain for a reader of the paper, so it is worth bringing them up in the discussion as future directions.

As requested we have added a single new paragraph to the Discussion that reads as follows:

“Our goal has been to produce a simple, tractable scenario for studying the effects of artificial selection on collectives, which while theoretical, is firmly connected to plausible biological experiments. The model could be extended in multiple ways in order to analyse the effects of additional factors, including impact of non-overlapping generations and variation in the timing of reproduction (which would introduce an element of bet-hedging), of migration and mixing between collectives (which could be akin to gamete production and zygote formation), and inclusion of more than two kinds of particle types. More complex selective regimes can also be envisaged, such as those that reward collectives based on absolute population size of particle types, which would allow less abstract collective function to be considered. However, regardless of these refinements, we suspect that our core conclusion will stand firm: collective-level selection favours particle dynamics that improve collective-level heredity. The ability to reliably re-establish successful adult states of past-generations from simpler and potentially noisy initial conditions is adaptive at the collective level.”